# Distributed quantum computing across an optical network link

D. Main[1✉], P. Drmota[1], D. P. Nadlinger[1], E. M. Ainley[1], A. Agrawal[1], B. C. Nichol[1], R. Srinivas[1], G. Araneda[1] & D. M. Lucas[1]

Distributed quantum computing (DQC) combines the computing power of multiple networked quantum processing modules, ideally enabling the execution of large quantum circuits without compromising performance or qubit connectivity[1,2]. Photonic networks are well suited as a versatile and reconfigurable interconnect layer for DQC; remote entanglement shared between matter qubits across the network enables all-to-all logical connectivity through quantum gate teleportation (QGT)[3,4]. For a scalable DQC architecture, the QGT implementation must be deterministic and repeatable; until now, no demonstration has satisfied these requirements. Here we experimentally demonstrate the distribution of quantum computations between two photonically interconnected trapped-ion modules. The modules, separated by about two metres, each contain dedicated network and circuit qubits. By using heralded remote entanglement between the network qubits, we deterministically teleport a controlled-Z (CZ) gate between two circuit qubits in separate modules, achieving 86% fidelity. We then execute Grover's search algorithm[5]—to our knowledge, the first implementation of a distributed quantum algorithm comprising several non-local two-qubit gates—and measure a 71% success rate. Furthermore, we implement distributed iSWAP and SWAP circuits, compiled with two and three instances of QGT, respectively, demonstrating the ability to distribute arbitrary two-qubit operations[6]. As photons can be interfaced with a variety of systems, the versatile DQC architecture demonstrated here provides a viable pathway towards large-scale quantum computing for a range of physical platforms.

The potential of quantum computing to revolutionize various fields ranging from cryptography to drug discovery is widely recognized[7,8]. However, regardless of the physical platform used to realize a quantum computer, scaling up the number of qubits while maintaining precise control and interconnectivity is a substantial technical challenge[9–11]. The distributed quantum computing (DQC) architecture, shown in Fig. 1, addresses this challenge by enabling large quantum computations to be executed by a network of quantum processing modules[1,2]. The modules each host a relatively small number of qubits and are interconnected through both classical and quantum information channels. By preserving the reduced complexity of the individual modules and transforming the scaling challenge into the task of building more modules and establishing an interface between them, the DQC architecture provides a scalable approach to fault-tolerant quantum computing[3,4].

The interface between modules could be realized by directly transferring quantum information between modules. However, losses in the interconnecting quantum channels would lead to the unrecoverable loss of quantum information. Quantum teleportation offers a lossless alternative interface, using only bipartite entanglement (for example, Bell states) shared between modules, together with local operations and classical communication to effectively replace the direct transfer of quantum information across quantum channels[12,13]. Quantum gate teleportation (QGT) efficiently implements non-local entangling gates between qubits in separate modules, consuming only one Bell pair and the exchange of two classical bits[14,15], as shown in Fig. 1b. Given arbitrary single-qubit and two-qubit operations within each node, QGT completes a universal gate set for the distributed quantum computer[13]. The primary advantage of teleportation-based schemes over direct transfer is the exclusive use of the quantum channel for generating identical Bell states; channel losses can be overcome by repetition without losing quantum information, and the distance between modules can be increased by inserting quantum repeaters[16]. Furthermore, channel noise may be suppressed using entanglement purification[17]. Because teleportation protocols are executed strictly after entanglement has been established, they enable continuous deterministic operation even if the entanglement is generated non-deterministically. This deterministic nature is crucial for scalability, eliminating the need for post-selection of singular successful outcomes out of an exponentially large set of undesired ones.

Teleportation protocols are agnostic to the physical implementation of the quantum channels, making them a versatile tool for DQC across different platforms. In the trapped-ion quantum charge-coupled device (QCCD) architecture, qubits can be dynamically transported between modules within a single chip[18]—or even across chips[19]—and thus be

[1]Clarendon Laboratory, Department of Physics, University of Oxford, Oxford, UK. ✉e-mail: dougal.main@physics.ox.ac.uk

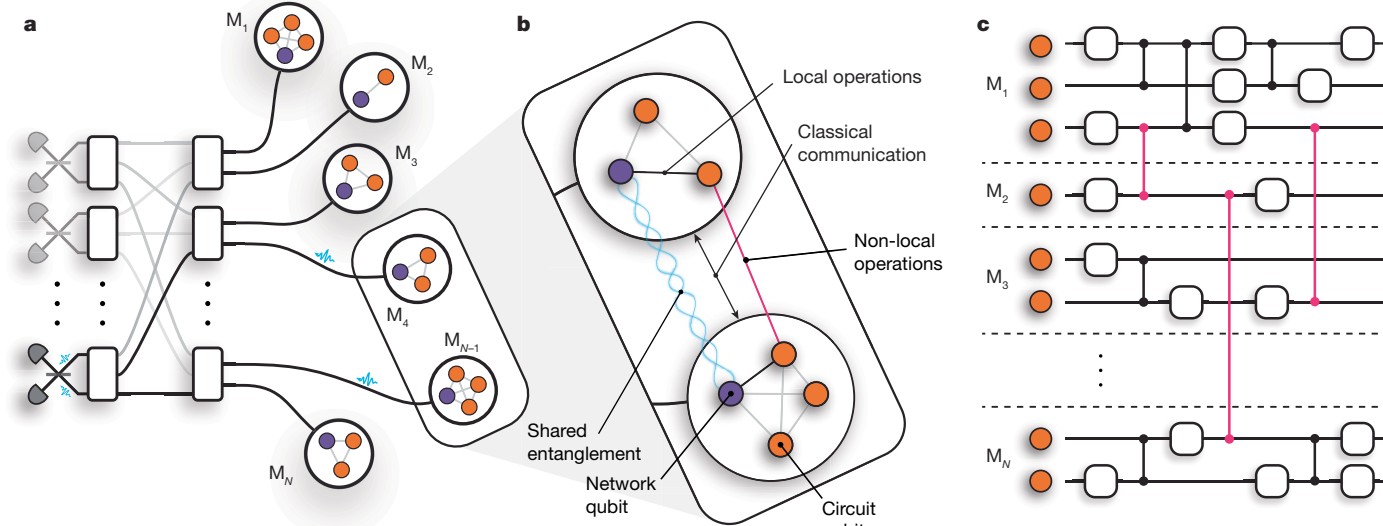

**Fig. 1 | DQC architecture. a**, Schematic of a DQC architecture comprising photonically interconnected modules. Entanglement is heralded between network qubits through the interference of photons on beam splitters. A photonic switchboard provides a flexible and reconfigurable network topology. **b**, The modules consist of at least one network qubit (purple) and at least one circuit qubit (orange), which may directly interact by means of local operations. QGT mediates non-local gate interactions (pink) between circuit qubits in separate modules. These protocols require the resources of shared entanglement, local operations and classical communication. **c**, A quantum circuit distributed across a network of small quantum processing modules that function together as a single, fully connected quantum computer.

used to mediate entangling gates between different trap zones[20,21]. Photons, however, make natural carriers of quantum information, as they can travel long distances without substantial degradation of their quantum state. Photonic interconnects enable all-to-all connectivity between qubits distributed across the network, whose topology can be dynamically reconfigured without the need to open up complex vacuum and/or cryogenic systems. Moreover, optical components are widely available and can be operated under ambient conditions. These properties make photonic interconnects particularly appealing for networking quantum computing modules, as shown in Fig. 1a. As shown in Fig. 1b, we consider modules containing 'network' and 'circuit' qubits with full interconnectivity through local quantum operations. Remote entanglement of network qubits in separate modules is generated by the interference of photons, in which reconfigurability and flexibility could be provided by means of a photonic switchboard. This entanglement can then be used to mediate multiqubit gates between the circuit qubits in different modules through QGT, enabling the network to function as a single, fully connected quantum processor, as shown in Fig. 1c. Quantum circuits can be partitioned freely in this architecture, down to a minimum of one circuit qubit per module in the fully distributed case. Heralded entanglement between spatially separated qubits has been achieved experimentally in a variety of platforms, including diamond colour centres[22,23], superconducting qubits[24], neutral atoms[25,26] and trapped ions[27–29].

QGT has been implemented probabilistically in purely photonic systems, requiring passive optical elements and post-selection to perform the conditional rotations that complete the gate teleportation[30,31]. Chou et al.[32] demonstrated deterministic teleportation of a controlled-NOT gate between two qubits encoded in the modes of two superconducting cavities on the same device, separated by about 2 cm, whereas a third cavity enabled the deterministic generation of entanglement between two transmon network qubits. Recently, there have been demonstrations of QGT between superconducting qubits within a single device, demonstrating the viability of QGT to overcome nearest-neighbour constraints in this architecture[33,34]. In the trapped-ion QCCD architecture, Wan et al.[20] demonstrated QGT between ions in two zones of the same trap, separated by about 840 μm; the entanglement was deterministically generated between qubits through local operations

before the qubits were transported. Furthermore, there have been demonstrations of heralded non-local entangling gates across a photonic quantum network in which photons are used to directly transfer quantum information between modules[35,36]. However, in these demonstrations, unavoidable photon loss destroyed the states of the circuit qubits, rendering these schemes non-deterministic. Until now, there has been (1) no demonstration of deterministic QGT across a quantum network and (2) no demonstration of distributed circuits comprising several non-local entangling gates. In photonic platforms, this has been prevented by the inability to store the photons between interactions[30,31], whereas in the QCCD demonstration, this was limited by the decoherence of the circuit qubits during the generation of entanglement[20].

In this work, we present, to our knowledge, the first demonstration of DQC across a network of two trapped-ion modules, each containing a network qubit and a circuit qubit, and separated by a macroscopic distance (about 2 m). We mediate deterministic two-qubit CZ interactions between the circuit qubits through QGT, using entanglement previously established across the network between the two network qubits. By making use of the robust storage of quantum information in the circuit qubits while generating subsequent rounds of entanglement between network qubits[37], we execute distributed quantum circuits comprising several non-local two-qubit gates. We demonstrate the distributed iSWAP and SWAP gates, which consist of two and three instances of QGT, respectively. The actions of all teleported gates are characterized using quantum process tomography. Finally, we implement Grover's algorithm on our distributed quantum computer.

## Teleportation of a CZ gate

Our apparatus, shown in Fig. 2a, consists of two trapped-ion modules, Alice and Bob, each co-trapping one [88]Sr[+] ion and one [43]Ca[+] ion (Methods). The Ca[+] ion provides a magnetic-field-insensitive 'circuit' qubit, $\mathcal{Q}_C := \{|0_C\rangle \equiv |F = 4, m_F = 0\rangle, |1_C\rangle \equiv |F = 3, m_F = 0\rangle\}$, in the ground hyperfine manifold, which has been used to demonstrate state-of-the-art quantum logic[38,39]. The Sr[+] ion, on the other hand, provides an efficient interface to the optical quantum network[28]. We define the

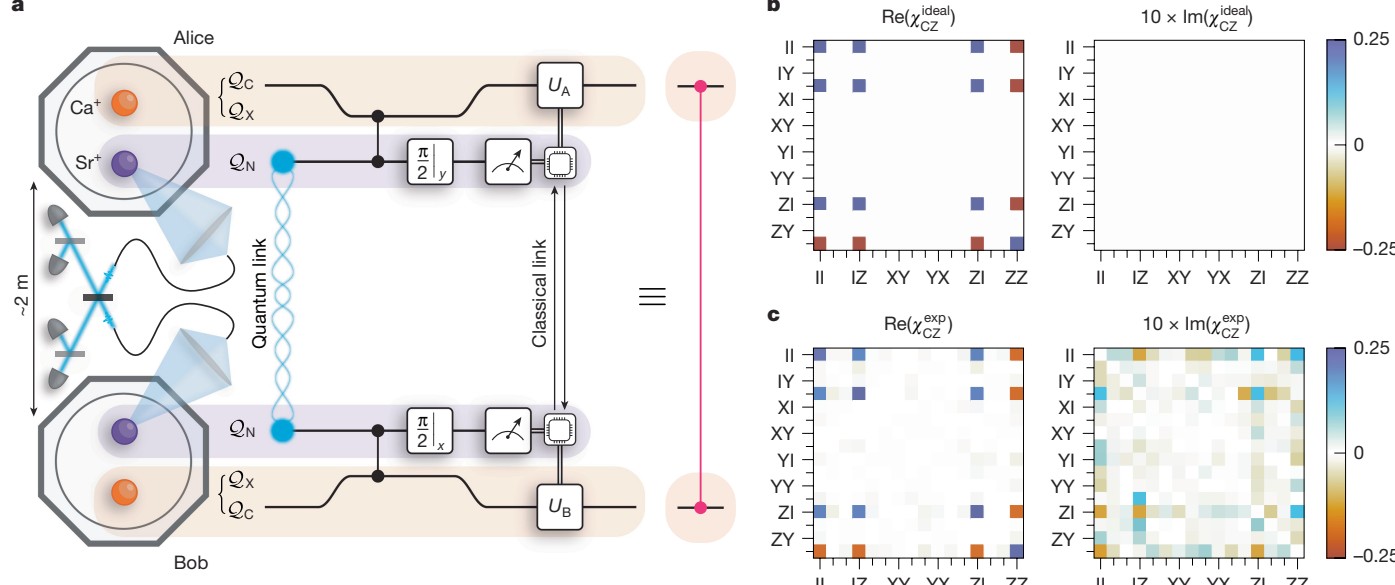

**Fig. 2 | Teleportation of a CZ gate between two trapped-ion modules.**
**a**, The two modules, Alice and Bob, each hold a $^{88}\text{Sr}^+$ ion (purple) and a $^{43}\text{Ca}^+$ ion (orange). $\text{Sr}^+$ provides a network qubit, $\mathcal{Q}_\text{N}$, whereas $\text{Ca}^+$ provides both a long-lived circuit qubit, $\mathcal{Q}_\text{C}$, and an auxiliary qubit, $\mathcal{Q}_\text{X}$. Before the protocol, the circuit qubits are in some arbitrary state. The protocol begins by generating entanglement between the network qubits through a photonic link. On heralding entanglement, each module applies a local CZ gate between the network and circuit qubits, using the auxiliary qubit temporarily to mediate

the gate mechanism. The outcomes of mid-circuit parity measurements of the network qubits are exchanged in real time through a classical (TTL) link connecting the control systems of the two modules. This information is used to condition local feed-forward operations, $U_\text{A}$ and $U_\text{B}$, on the circuit qubits—completing the teleportation of the CZ gate. **b**, Process matrix for an ideal CZ gate. **c**, Measured process matrix, reconstructed through quantum process tomography, yielding an average gate fidelity of 86.2(9)% compared with an ideal CZ gate.

network qubit in $\text{Sr}^+$ by $\mathcal{Q}_\text{N} := \{|0_\text{N}\rangle \equiv |\text{S}_{1/2}, m_J = -\tfrac{1}{2}\rangle, |1_\text{N}\rangle \equiv |\text{D}_{5/2}, m_J = -\tfrac{3}{2}\rangle\}$. To implement local entangling operations between these two species, we use the light-shift gate mechanism[40] between $\mathcal{Q}_\text{N}$ and an auxiliary qubit in the ground hyperfine manifold of $\text{Ca}^+$, $\mathcal{Q}_\text{X} := \{|0_\text{X}\rangle \equiv |F = 4, m_F = +4\rangle, |1_\text{X}\rangle \equiv |F = 3, m_F = +3\rangle\}$, which—unlike the $\mathcal{Q}_\text{C}$ qubit—experiences the necessary light shifts (Methods). At the points at which we want to perform the local entangling gate, we transfer the quantum information stored in $\mathcal{Q}_\text{C}$ temporarily to $\mathcal{Q}_\text{X}$ to perform the gate operations (Methods).

The QGT protocol used here to mediate CZ gates between the circuit qubits in separate modules is shown in Fig. 2. We allow the circuit qubits to start in an arbitrary state $|\psi_\text{in}^\text{AB}\rangle \in \mathcal{Q}_\text{C}^{\otimes 2}$, which could be part of a larger, long-running computation. We begin the QGT protocol by generating the remotely entangled Bell state

$$|\psi^+\rangle = \frac{|10\rangle + |01\rangle}{\sqrt{2}} \in \mathcal{Q}_\text{N}^{\otimes 2},$$

between the network qubits[28], with a fidelity of 96.89(8)% (Methods). This entanglement is generated through a try-until-success process, for which a herald indicates a success. In contrast to the network qubits, the circuit qubits provide a robust quantum memory[37], enabling storage of the encoded quantum information until the remote entanglement is successfully heralded. At this stage, we map the state stored in the circuit qubits ($\mathcal{Q}_\text{C}$) to the auxiliary qubits ($\mathcal{Q}_\text{X}$) in preparation for the local entangling operations (Methods). In each module, we perform local CZ gates between the network and auxiliary qubits (Methods), before transferring the auxiliary qubit back to the circuit qubit. We then perform mid-circuit measurements of the network qubits in the $X$ and $Y$ bases in Alice and Bob, respectively. The modules exchange the measurement outcomes in real time—using a classical (TTL) link between their control systems—and perform single-qubit feed-forward operations conditioned on the exchanged bits to complete the gate

teleportation protocol (Methods). This implements the non-local gate $|\psi_\text{in}^\text{AB}\rangle \to U_\text{CZ}^\text{AB}|\psi_\text{in}^\text{AB}\rangle$.

We characterize the QGT protocol using quantum process tomography (Methods) to reconstruct the process matrix, $\chi_\text{CZ}^\text{exp}$, providing a complete description of the action of the teleported CZ gate on the two circuit qubits. Compared with the ideal CZ process, shown in Fig. 2b, the reconstructed process matrix for the teleported gate, shown in Fig. 2c, has an average gate fidelity of 86.2(9)%. The QGT protocol is completely self-contained—the input states of the circuit qubits are set before the execution of the non-local gate—and output states are available for further computation. With single-qubit rotations of the circuit qubits, this teleported CZ gate is a key element of a gate set for DQC, enabling the modules to act as a single, fully connected universal quantum processor.

## DQC

In general, any arbitrary two-qubit unitary operation can be decomposed into at most three CZ gates[6]. We demonstrate our ability to perform sequential rounds of QGT by executing the CZ decompositions of the iSWAP and SWAP gates, shown in (1) in Fig. 3a,b, comprising two and three instances of QGT, respectively. As with the teleported CZ gate, we characterize these circuits through quantum process tomography (Methods); see (2) in Fig. 3a,b. From the reconstructed process matrices, we measure average gate fidelities of 70(2)% and 64(2)% for the iSWAP and SWAP gates, respectively. By constructing circuits with several instances of QGT—enabled by our ability to perform QGT deterministically and on demand—we demonstrate the ability to perform universal DQC.

Finally, we implement Grover's algorithm[5,41,42] on our distributed quantum processor. This algorithm searches through a set of unsorted items, $x \in L$, to find a particular item, $a \in L$. The search problem is represented by the function

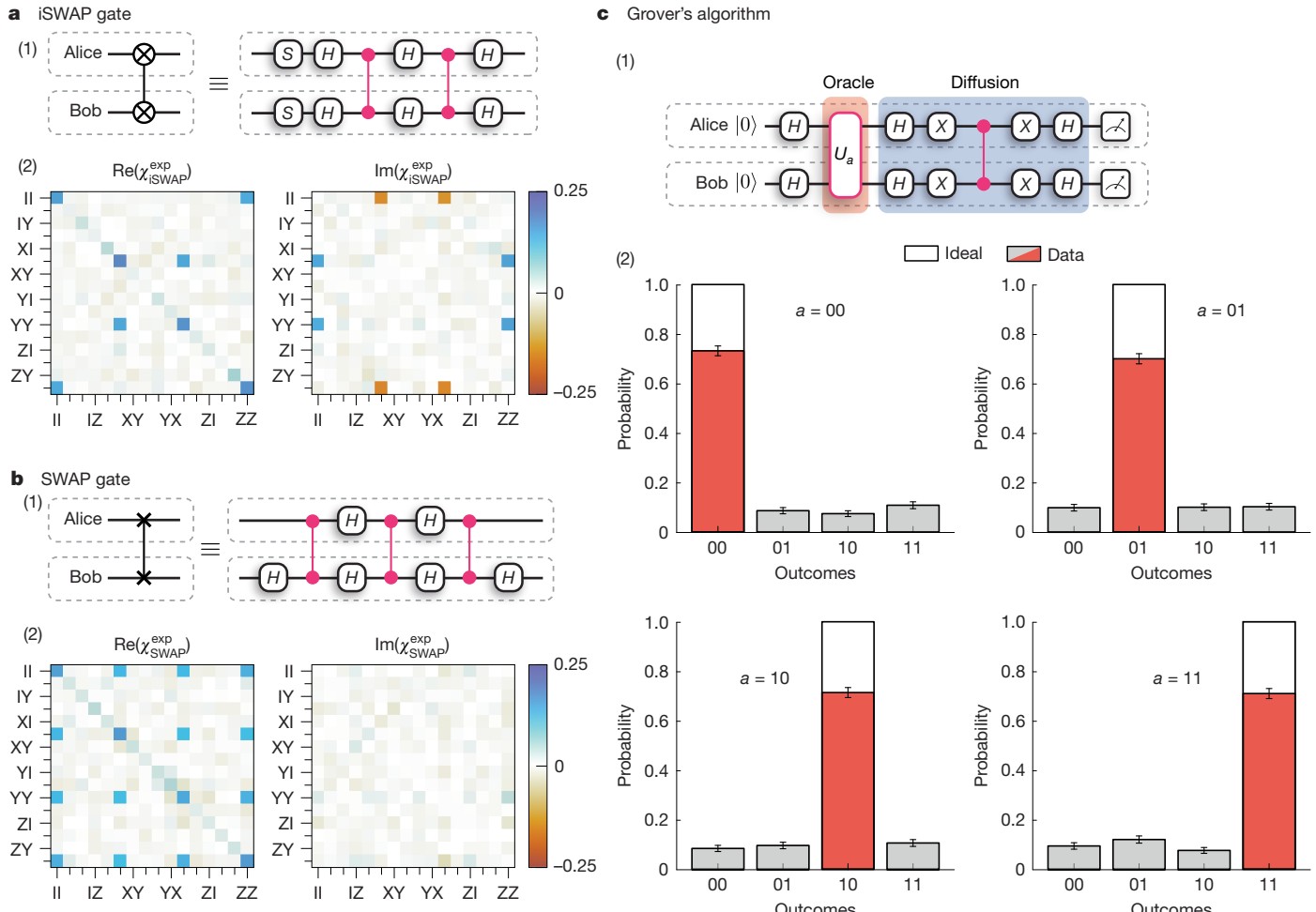

**a** iSWAP gate

**b** SWAP gate

**c** Grover's algorithm

**Fig. 3 | DQC results. a,b,** (1) CZ decompositions of the distributed iSWAP (**a**) and SWAP (**b**) circuits, comprising two and three instances of QGT, respectively. (2) The reconstructed process matrices for the iSWAP (**a**) and SWAP (**b**) gates indicate average gate fidelities of 70(2)% and 64(2)%, respectively. **c**, Grover's algorithm. (1) Circuit comprising two instances of QGT: the first implements the

Grover oracle call, which marks a particular state, $a$, and the second implements the diffusion circuit. (2) Measurement outcomes from 500 repetitions of Grover's algorithm per marked state; the average success probability is 71(1)%. All error bars indicate one standard deviation.

$$f_a(x) = \begin{cases} 1 & \text{if } x = a, \\ 0 & \text{otherwise.} \end{cases}$$

In the two-qubit case, there are four items to search through. Classically, the item $a$ could be identified with, on average, two queries of the function $f_a(x)$. Using the quantum circuit shown in (1) in Fig. 3c, the same task can be accomplished with only one query. After preparing a superposition of all possible inputs with parallel Hadamard gates, an instance of QGT implements the oracle, which performs the mapping $U_a : |x\rangle \rightarrow (-1)^{f_a(x)}|x\rangle$, marking the state $|a\rangle$. A second instance of QGT implements the Grover diffusion circuit, which decodes the quantum information provided by the oracle into an observable. In the two-qubit case considered here, the application of the Grover diffusion circuit should leave the register in the state $|a\rangle$, which is the solution to the function $f_a$, and thus a measurement of the register yields the solution to the search problem with unit probability. In the case of $N$ items, to approach unit probability of obtaining the solution, we would require $\mathcal{O}(\sqrt{N})$ iterations of the oracle–diffusion circuit, compared with $\mathcal{O}(N/2)$ for a classical search.

The results of Grover's algorithm—executed on our distributed quantum processor—are shown in (2) in Fig. 3c. For the marked states $a \in \{00, 01, 10, 11\}$, we obtain the correct result with an average success rate of 71(1)%. To our knowledge, this represents the first deterministic execution of any algorithm on a distributed quantum computer.

## Discussion

The performance of our distributed quantum circuits is consistent with the errors from the teleported CZ gates. We summarize the leading error sources affecting our teleported CZ gate in Table 1. The measured fidelity of our gate is slightly lower than that predicted by the error budget, which we attribute to drifts in the calibration of various components over the duration of the data acquisition. Most of the identified errors occur during local operations in each module. Our local errors do not represent the state of the art for trapped-ion processors; however, local operations exceeding the approximately 99% fidelity threshold for fault-tolerant quantum computing have been demonstrated in this platform[39,40,43–48]. Relevant to our implementation, Hughes et al.[40] demonstrated mixed-species two-qubit gates between $^{88}Sr^+$ and $^{43}Ca^+$ ions with a gate fidelity of 99.8(1)%. We therefore conclude that the technical limitations in our implementation can be overcome. The other notable source of error is the remote entanglement of the network qubits across the photonic quantum network; we observe a fidelity of the remotely entangled network qubits to the desired $|\Psi^+\rangle$ state of 96.89(8)%. Unlike the local operations, the performance of our remote entanglement is at the state of the art. To improve this, and hence enable the teleportation of high-fidelity entangling gates between modules, entanglement purification could be used to distribute high-fidelity entangled states from several lower-fidelity entangled states[17,49].

**Table 1 | Error budget for CZ gate teleportation**

| Source | Error | |
|---|---|---|
| | Alice | Bob |
| Raw entanglement | 3.11(8)% | |
| Mixed-species gate | 2.4(2)% | 2.0(2)% |
| $\mathcal{Q}_C$ decoherence | 1.9(4)% | 1.8(5)% |
| $\mathcal{Q}_X \leftrightarrow \mathcal{Q}_C$ transfer | 0.76(3)% | 0.52(1)% |
| Mid-circuit measurement | 0.091(3)% | 0.122(2)% |
| $\mathcal{Q}_C$ rotations | 0.016(1)% | 0.015(1)% |
| Predicted total error | 12.1(6)% | |

The characterization of each error contribution is discussed in Methods.

Our implementation features a single circuit qubit in each module; however processors with larger numbers of qubits have been realized. With three circuit qubits (and one network qubit) per module, the purification of arbitrary quantum channels would be possible[49]. The capabilities of the individual modules may be extended even further by deploying the QCCD architecture. With recent demonstrations in both academic research[50] and industry[21] highlighting the power of this approach, embedding these systems in a quantum network would combine their power with the reconfigurability and flexibility of the DQC architecture. Conversely, computational bottlenecks associated with ion transport overheads observed in the QCCD architecture[21] could be mitigated using photonic interconnects integrated into a single device[51].

Although the results presented here were achieved using trapped-ion quantum processing modules, photons may be interfaced with a variety of systems. The connectivity and reconfigurability enabled by photonic networks provides a scalable approach for other quantum computing platforms, such as diamond colour centres and neutral atoms. Also, modules of different platforms could be connected by means of wavelength conversion, enabling a hybrid DQC platform. Furthermore, teleportation protocols are not limited to qubits; they can be extended to higher-dimensional quantum computing models, such as qudits[52] and continuous-variable quantum computing[53,54], allowing these platforms to benefit from the DQC architecture. Quantum repeater technology[16] would enable large physical separation between the quantum processing modules, thereby paving the way for the development of a quantum internet[55]. The scope of these networks extends beyond quantum computing technologies; the ability to control distributed quantum systems, as enabled by this architecture, to engineer complex quantum resources has applications in multipartite secret sharing[56], metrology[57] and examining fundamental physics[58].

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

## Methods

### Dual-species ion-trap modules

Our apparatus comprises two trapped-ion processing modules, Alice and Bob. Each module, shown in Extended Data Fig. 1, consists of an ultrahigh-vacuum chamber containing a room-temperature, microfabricated surface Paul trap; the trap used in Alice (Bob) is a HOA-2 (ref. 59) (Phoenix[60]) trap, fabricated by Sandia National Laboratories. In each module, we co-trap $^{88}Sr^+$ and $^{43}Ca^+$ ions. Each species of ion is addressed by means of a set of lasers used for cooling, state preparation and readout. A high-numerical-aperture (0.6 NA) lens enables single-photon collection from the $Sr^+$ ions. A roughly 0.5 mT magnetic field is applied parallel to the surface of the trap to lift the degeneracies of the Zeeman states and provide a quantization axis.

As outlined in the main text, the $Sr^+$ ion provides an optical network qubit, $Q_N$, which is manipulated directly using a 674-nm laser. The ground hyperfine manifold of the $Ca^+$ ion provides a circuit qubit, $Q_C$. At about 0.5 mT, the sensitivity of the $Q_C$ qubit transition frequency to magnetic-field fluctuations is 122 kHz mT$^{-1}$, that is, about two orders of magnitude lower than that of the $Q_N$ qubit with a sensitivity of $-11.2$ MHz mT$^{-1}$, making it an excellent memory for quantum information[37]. Furthermore, we define an auxiliary qubit, $Q_X$, in the ground hyperfine manifold of $Ca^+$ for implementing local entangling operations, state preparation and readout. The measured state-preparation and measurement errors for each qubit are presented in Extended Data Table 1.

The spectral isolation between the two species allows us to address one species without causing decoherence of the quantum information encoded in the other species. We make use of this property for sympathetic cooling, mid-circuit measurement and interfacing with the quantum network during circuits.

### Quantum process tomography

The action of a quantum process acting on a system of $N$ qubits may be represented by the process matrix $\chi_{\alpha\beta}$ such that

$$\mathcal{E}(\rho) = \sum_{\alpha,\beta=0}^{D-1} \chi_{\alpha\beta} P_\alpha \rho P_\beta^\dagger, \tag{1}$$

in which $P_\alpha \in \mathcal{P}^{\otimes N}$ are the set of single-qubit Pauli operators $\mathcal{P} = \{\mathbb{I}, \sigma_x, \sigma_y, \sigma_z\}$ and $D = \dim(\mathcal{P}^{\otimes N}) = 4^N$. Quantum process tomography enables us to reconstruct the matrix $\chi_{\alpha\beta}$, thereby attaining a complete characterization of the process.

Quantum process tomography is performed by preparing the qubits in the states $\rho_i = |\psi_i\rangle\langle\psi_i|$, in which $|\psi_i\rangle$ are chosen from a tomographically complete set

$$|\psi_i\rangle \in \left\{ |0\rangle, |1\rangle, \frac{|0\rangle + |1\rangle}{\sqrt{2}}, \frac{|0\rangle + i|1\rangle}{\sqrt{2}} \right\}, \tag{2}$$

performing the process, $\mathcal{E}$, followed by measuring the output state $\mathcal{E}[\rho_i]$ in a basis chosen from a tomographically complete set. Using diluted maximum-likelihood estimation[61], the outcomes of the measurements can be used to reconstruct the $\chi$ matrix representing the process. In practice, the input states are created by rotating $|0\rangle$ to $|\psi_i\rangle = R_i|0\rangle$ with

$$R_i \in \left\{ \mathbb{I}, \sigma_x, \frac{1}{\sqrt{2}}(\mathbb{I} - i\sigma_y), \frac{1}{\sqrt{2}}(\mathbb{I} + i\sigma_x) \right\}. \tag{3}$$

Likewise, the tomographic measurements are performed by rotating the output state $\mathcal{E}[\rho_i]$ by $R_j^\dagger$ (equation (3)) and subsequently measuring it in the $\sigma_z$ basis. Ideally, this sequence implements the projectors $P_{0,j} = |\psi_j\rangle\langle\psi_j|$ and $P_{1,j} = |\psi_{\perp,j}\rangle\langle\psi_{\perp,j}|$, in which $\langle\psi_{\perp,j}|\psi_j\rangle = 0$.

However, state-preparation and measurement errors would manifest as errors in the reconstructed process. We therefore model the imperfect state preparation by replacing the ideal input states, $|\psi_i\rangle$, with the states

$$\rho_i = R_i[(1-\epsilon)|0\rangle\langle0| + \epsilon|1\rangle\langle1|]R_i^\dagger, \tag{4}$$

in which $\epsilon$ is the state-preparation error. Note that this model assumes that imperfect state preparation leaves the ionic state within the qubit subspace; however, imperfect state preparation often results in leakage outside this subspace. Nevertheless, for the purposes of our analysis, this model is sufficient.

Similarly, we model the imperfect qubit readout by replacing the projectors $P_{0,j}$ and $P_{1,j}$ with the positive-operator-valued measures

$$M_{0,j} = (1 - \epsilon_0)R_j^\dagger|0\rangle\langle0|R_j + \epsilon_1 R_j^\dagger|1\rangle\langle1|R_j \tag{5}$$

$$M_{1,j} = (1 - \epsilon_1)R_j^\dagger|1\rangle\langle1|R_j + \epsilon_0 R_j^\dagger|0\rangle\langle0|R_j, \tag{6}$$

in which $\epsilon_0$ and $\epsilon_1$ are the computational basis readout errors. The values for these errors are given in Extended Data Table 1.

To quantify the performance of a process, $\mathcal{E}$, compared with an ideal unitary process, $U$, we make use of the average gate fidelity

$$\bar{F}_{\mathcal{E},U} = \int d\psi \langle\psi|U^\dagger \mathcal{E}(|\psi\rangle\langle\psi|)U|\psi\rangle \tag{7}$$

as defined by Nielsen[62], which corresponds to the fidelity averaged over all pure input states. We define the process $\mathcal{E}'$ as the application of the process $\mathcal{E}$ followed by the inverse of the ideal process $U$, such that

$$\mathcal{E}'(\rho) = U^\dagger \mathcal{E}(\rho)U. \tag{8}$$

If $\chi'_{\alpha\beta}$ is the process matrix representing $\mathcal{E}'$, as in equation (1), then the average gate fidelity can be expressed as

$$\bar{F}_{\mathcal{E},U} = \frac{1 + d\chi'_{00}}{1 + d}, \tag{9}$$

in which $d$ is the dimension of the Hilbert space.

Resampling of the measurement outcomes is used to generate new datasets, which are analysed in the same way as the original dataset and are used to determine the sensitivity of the analysis to the statistical fluctuations in the input data. The error bar on the average gate fidelity of a reconstructed process is quoted as the standard deviation of average gate fidelities of processes reconstructed from resampled datasets.

### Remote entanglement generation

The heralded generation of remote entanglement between network qubits in separate modules, outlined in ref. 28, is central to our QGT protocol. Spontaneously emitted 422-nm photons entangled with the $Sr^+$ ions are collected from each module using high-numerical-aperture lenses and single-mode optical fibres bring the photons to a central Bell-state analyser, in which a measurement of the photons projects the ions into a maximally entangled state[28,63,64]. This forms the photonic quantum channel interconnecting the two modules. Following ref. 28, we use a 674-nm π-pulse to map the remote entanglement from the ground-state Zeeman qubit to an optical qubit, which we refer to as the network qubit, to minimize the number of quadrupole pulses in subsequent operations. Successful generation of entanglement is heralded by particular detector click patterns and, after subsequent local rotations, indicate the creation of the maximally entangled $\Psi^+$ Bell state

$$|\Psi^+\rangle = \frac{|10\rangle + |01\rangle}{\sqrt{2}} \in \mathcal{Q}_N^{\otimes 2}.$$

This process is executed while simultaneously storing quantum information in the circuit qubits, which—as demonstrated in ref. 37—are robust to this network activity.

Each entanglement generation attempt takes 1,168 ns and it takes 7,084 attempts to successfully herald entanglement on average, corresponding to a success probability of $1.41 \times 10^{-4}$. To mitigate heating of the ion crystal, we interleave 200 μs of entanglement generation attempts with 2.254 ms of sympathetic recooling of the $Sr^+$–$Ca^+$ crystal using the $Sr^+$ ion. The sympathetic recooling comprises 1.254 ms of Doppler cooling, followed by 1 ms of electromagnetically induced transparency cooling. Overall, this results in an average entanglement generation rate of $9.7~s^{-1}$ (equivalently, it takes, on average, 103 ms to generate entanglement between network qubits), although this rate could be increased by optimizing the interleaved cooling sequence. This rate is lower than the $182~s^{-1}$ rate previously reported in our apparatus[28] owing to the extra cooling. We characterize the remote entanglement using quantum state tomography; by performing tomographic measurements on $2 \times 10^5$ copies of the remotely entangled state, we reconstruct the density matrix of the network qubits, $\rho_N^{AB}$, shown in Extended Data Fig. 2d. To isolate the fidelity of the 'quantum link' in Fig. 2, we account for the imperfect tomographic measurements in the reconstruction of the density matrix using the positive-operator-valued measures in equations (5) and (6). The fidelity of the reconstructed state to the desired $\Psi^+$ Bell state, given by $\langle \Psi^+ | \rho_N^{AB} | \Psi^+ \rangle$, is 96.89(8)%.

We believe that the fidelity is predominantly limited by errors occurring during the generation of ion–photon entanglement in each module, rather than imperfections in the apparatus used to perform the projective Bell-state measurement. In particular, we attribute the primary sources of error to polarization mixing due to imperfections in the imaging systems used to collect single photons from each module and to drifts in the birefringence of the optical fibres that form the network link between the modules.

## Circuit qubit memory during entanglement generation

Because each instance of QGT requires the generation of entanglement between network qubits, it is necessary to ensure that the circuit qubits preserve their encoded quantum information during this process. Owing to their low sensitivity to magnetic-field fluctuations, the circuit qubits have exhibited roughly 100 ms coherence times and, in previous work, we demonstrated these qubits to be robust to network activity[37]. We further suppress dephasing through dynamical decoupling. Typically, dynamical decoupling is implemented over a fixed period of time; however, the success of the entanglement generation process is non-deterministic and would therefore leave the dynamical decoupling sequence incomplete.

One solution would be to complete the dynamical decoupling pulse sequence once the entanglement has been generated. However, it is desirable to minimize the time between heralding the entanglement generation and performing the QGT protocol, to prevent dephasing of the network qubits. Instead, we make use of the fact that the action of a dynamical decoupling pulse on one of the circuit qubits can be propagated through the teleported CZ gate as

$$(X \otimes I)U_{CZ} = U_{CZ}(X \otimes Z). \tag{10}$$

We therefore perform the dynamical decoupling pulses on the circuit qubits until we obtain a herald of remote entanglement, at which point we immediately perform the QGT sequence—implementing a CZ gate on the state of the circuit qubits at the point of interruption. Once this gate is completed, we perform the remaining dynamical decoupling pulses (without any interpulse delay) and use equation (10) to apply

the appropriate $Z$ rotations required to correct for the propagation through the CZ gate. With this method, we suppress the dephasing errors in the circuit qubits during entanglement generation while minimizing the time between successfully heralding the entanglement and consuming it for QGT.

We deploy Knill dynamical decoupling[65,66] with a 7.4-ms interpulse delay (corresponding to a pulse every three rounds of interleaved entanglement attempts and recooling). We use quantum process tomography to reconstruct the process of storing the quantum information while generating entanglement; ideally, this process would not alter the quantum information stored in the circuit qubit. Quantum process tomography is implemented by choosing input states for the circuit qubits from the tomographically complete set given in equation (2), generating remote entanglement between the network qubits while dynamically decoupling the circuit qubits, then—on successful herald—completing the dynamical decoupling sequence and performing tomographic measurements of the circuit qubits. The reconstructed process matrices for each module corresponding to the action of storing quantum information during entanglement generation are shown in Extended Data Fig. 2c. We observe fidelities to the ideal operation of 98.1(4)% and 98.2(5)% for Alice and Bob, respectively.

## Local mixed-species entangling gates

The ability to perform logical entangling gates between ions of different species allows us to separate the roles of network and circuit ions. We implement mixed-species entangling gates following the approach taken in ref. 40, in which geometric phase gates are deterministically executed using a single pair of 402-nm Raman beams, as shown in Extended Data Fig. 3. Here we apply the gate mechanism directly to the network qubit in $Sr^+$—rather than the Zeeman ground-state qubit, as done by Hughes et al.[40] and Drmota et al.[37]—at the cost of a slightly reduced gate efficiency that is compensated for by the use of higher laser powers. This enables us to perform mixed-species CZ gates between the network and auxiliary qubits. We characterize our mixed-species entangling gates using quantum process tomography in each module, reconstructing the process matrices $\chi_{CZ}$ representing the action of the local CZ gate acting between the network and auxiliary qubits. The reconstructed process matrices for each module are shown in Extended Data Fig. 3d. Compared with the ideal CZ gate, we observe average gate fidelities of 97.6(2)% and 98.0(2)% for Alice and Bob, respectively.

## Hyperfine qubit transfer

Because the circuit qubit does not participate in the mixed-species gate, the gate interaction is performed on the network and auxiliary qubits. Consequently, we require the ability to map coherently between the circuit and auxiliary qubit before and after the local operations. As shown in Extended Data Fig. 4, this mapping is performed using a pair of 402-nm Raman beams detuned by about 3.2 GHz, to coherently drive the transitions within the ground hyperfine manifold of $Ca^+$.

The transfer of the circuit qubit to the auxiliary qubit begins with the mapping of the state $|0_C\rangle$ to the state $|0_X\rangle$. However, owing to the near degeneracy of the transition $\mathcal{T}_0 : |0_C\rangle \leftrightarrow |F = 3, m_F = +1\rangle$ and the transition $\mathcal{T}_1 : |1_C\rangle \leftrightarrow |F = 4, m_F = +1\rangle$ (Extended Data Fig. 4), separated by only about 15 kHz, it is not possible to map the $|0_C\rangle$ state out of the circuit qubit without off-resonantly driving population out of the $|1_C\rangle$ state. We suppress this off-resonant excitation using a composite pulse sequence, shown in (1) in Extended Data Fig. 4b, comprising three pulses resonant with the $\mathcal{T}_0$ transition, with pulse durations equal to the $2\pi$ time of the $\mathcal{T}_1$ transition, and phases optimized to minimize the off-resonant excitation. This pulse sequence allows us to simultaneously perform a π-pulse on the $\mathcal{T}_0$ transition and the identity on the off-resonantly driven $\mathcal{T}_1$ transition. Raman π-pulses are then used to complete the mapping to the $|0_X\rangle$ state. Another sequence of Raman π-pulses coherently maps $|1_C\rangle \rightarrow |1_X\rangle$, thereby completing the transfer

of the circuit qubit to the auxiliary qubit, $\mathcal{Q}_C \to \mathcal{Q}_X$. To implement the mapping $\mathcal{Q}_X \to \mathcal{Q}_C$, the same pulse sequence is applied in reverse.

We characterize our $\mathcal{Q}_C \leftrightarrow \mathcal{Q}_X$ mapping sequence by performing a modification of single-qubit randomized benchmarking (RBM), in which we alternate Clifford operations on the $\mathcal{Q}_C$ and $\mathcal{Q}_X$ qubits, as illustrated in Extended Data Fig. 4c. We assume that (1) the single-qubit gate errors for the $\mathcal{Q}_C$ and $\mathcal{Q}_X$ qubits are negligible compared with the $\mathcal{Q}_C \leftrightarrow \mathcal{Q}_X$ transfer infidelity (we typically observe single-qubit gate errors of around $1 \times 10^{-4}$ for the $Ca^+$ hyperfine qubits) and (2) the fidelity of the transfer $\mathcal{Q}_C \to \mathcal{Q}_X$ is similar to $\mathcal{Q}_X \to \mathcal{Q}_C$. We therefore we model the survival probability as

$$S(m) = \frac{1}{2} + Bp^m$$

in which $m$ is the number of hyperfine transfers, $B$ accounts for state-preparation and measurement error offsets and $p$ is the depolarizing probability for the transfer, related to the error per transfer as

$$\epsilon_{C \leftrightarrow X} = \frac{1-p}{2}.$$

The RBM results are shown in Extended Data Fig. 4c; we measure an error per transfer of $3.8(2) \times 10^{-3}$ ($2.6(1) \times 10^{-3}$) for Alice (Bob).

## Conditional operations

To complete the QGT protocol, the two modules perform mid-circuit measurements of the network qubits, exchange the measurement outcomes and apply a local rotation of their circuit qubits conditioned on the outcomes of the measurements. By virtue of the spectral isolation between the two species of ions, mid-circuit measurements of the network qubits can be made without affecting the quantum state of the circuit qubits. The mid-circuit measurement outcomes, $m_A, m_B \in \{0, 1\}$, are exchanged in real time through a classical communication channel between the modules—in our demonstration, this is a TTL link connecting the control systems of the two modules. Following the exchange of the measurement outcomes, the modules, Alice and Bob, perform the conditional rotations $U_A$ and $U_B$, respectively, in which

$$U_A = \begin{cases} S^\dagger & \text{if } m_A \oplus m_B = 0, \\ S & \text{otherwise}, \end{cases}$$

$$U_B = \begin{cases} S & \text{if } m_A \oplus m_B = 0, \\ S^\dagger & \text{otherwise}, \end{cases}$$

in which $S = \text{diag}(1, i)$.

Errors in the mid-circuit measurements of the network qubits will result in the application of the wrong conditional rotation; effectively, this would appear as a joint phase flip of the circuit qubits following the teleported gate. The mid-circuit measurement errors arise from the non-ideal single-qubit rotation of the network qubit to map the measurement basis onto the computational basis and errors owing to the fluorescence detection of the network qubit. Using RBM, we measure single-qubit gate errors for the network qubits of $4.8(3) \times 10^{-4}$ and $9.8(3) \times 10^{-4}$ for Alice and Bob, respectively. The error in the fluorescence detection is estimated from the observed photon scattering rates of $\mathcal{Q}_N$ states, as well as the approximately 390 ms lifetime of the

$|1_N\rangle$ state[67]. We choose a mid-circuit measurement duration of 500 µs and estimate fluorescence detection errors of $6.6(1) \times 10^{-4}$ and $5.51(2) \times 10^{-4}$ for Alice and Bob, respectively. Combining these error mechanisms, we estimate contributions to the teleported CZ gate error of 0.091(3)% and 0.122(2)% for Alice and Bob, respectively.

## Data availability

The datasets generated during this study are available from D.M. and D.M.L. (email: david.lucas@physics.ox.ac.uk) on reasonable request.

## Code availability

All analysis code that supports the plots in this paper and other findings of this study are available from the corresponding author on reasonable request.

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

**Acknowledgements** We thank O. Băzăvan, S. Saner and D. Webb for maintenance of the 674-nm laser system; C. Ballance and L. Stephenson for the design and construction of the apparatus; J. Blackmore and P. Juhász for comments on the paper; Sandia National Laboratories for supplying the ion traps used in this experiment; and the developers of the control system ARTIQ[68]. D.M. acknowledges support from the US Army Research Office (ref. W911NF-18-1-0340). D.P.N. acknowledges support from Merton College, Oxford. E.M.A. acknowledges support from the UK EPSRC 'Quantum Communications Hub' EP/T001011/1. R.S. acknowledges funding from an EPSRC fellowship EP/W028026/1 and Balliol College, Oxford. G.A. acknowledges support from Wolfson College, Oxford. This work was supported by the UK EPSRC 'Quantum Computing and Simulation Hub' EP/T001062/1.

**Author contributions** D.M., P.D., D.P.N., E.M.A., A.A., B.C.N., R.S. and G.A. built and operated the experimental apparatus. D.M. led the experimental work, with assistance from P.D. and D.P.N. D.M. performed the data analysis and prepared the paper, with input from all authors. D.M.L. secured funding and supervised the work. All authors contributed to the discussion and interpretation of results.

**Competing interests** R.S. is partly employed by Oxford Ionics Ltd. The other authors declare no competing interests.

**Additional information**
**Correspondence and requests for materials** should be addressed to D. Main.

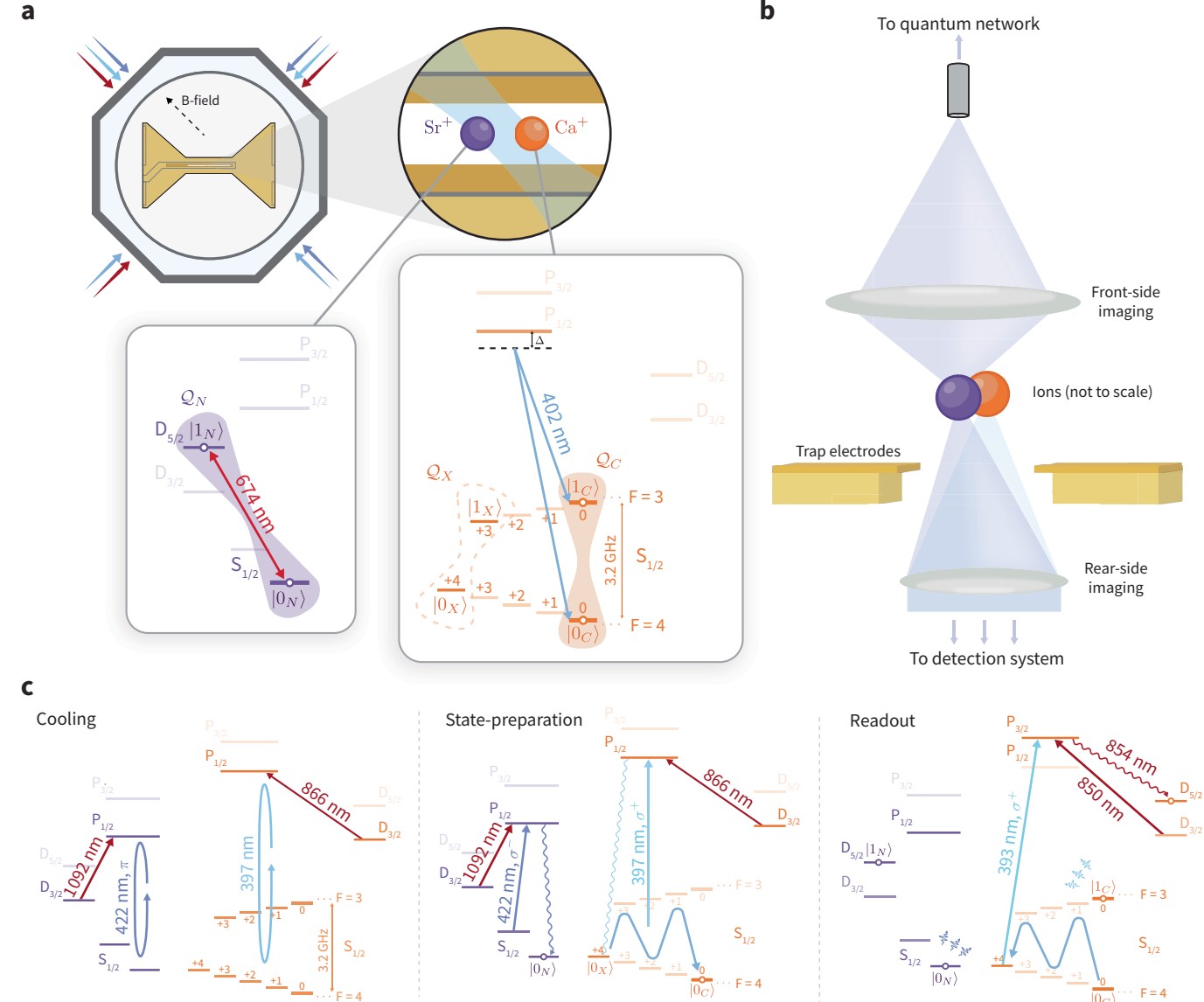

**Extended Data Fig. 1 | Outline of a trapped-ion module. a**, An ultrahigh-vacuum chamber houses a microfabricated surface Paul trap, which co-traps one $^{88}Sr^+$ ion and one $^{43}Ca^+$ ion. The ions are manipulated using lasers, which are delivered parallel to the surface of the trap. The $Sr^+$ ion provides an optical network qubit, $Q_N$, which is coherently manipulated using a 674-nm laser. The ground hyperfine manifold of the $Ca^+$ ion provides a circuit qubit, $Q_C$, and an auxiliary qubit, $Q_X$. The qubits in the ground hyperfine manifold are addressed using a pair of 402-nm Raman beams. **b**, The rear-side imaging system is used to perform fluorescence detection for qubit readout of both species. The front-side imaging system is used for single-photon collection from the $Sr^+$ ion during

the generation of entanglement. A high-numerical-aperture (0.6 NA) lens couples the single photons into a single-mode optical fibre, which connects to the optical quantum network. Both imaging systems are outside the vacuum chamber. **c**, Energy-level diagrams for cooling, state preparation and readout of each species. For state preparation and readout of the circuit qubit $Q_C$ in $Ca^+$, we use the auxiliary qubit $Q_X$. During state preparation, we prepare $|0_X\rangle$ through optical pumping and then transfer it to $|0_C\rangle$ using Raman $\pi$-pulses. For readout, we transfer $|0_C\rangle$ to $|0_X\rangle$ before shelving to the $D_{5/2}$ manifold. Fluorescence detection is then used for both species; ions in the shelved state do not scatter photons.

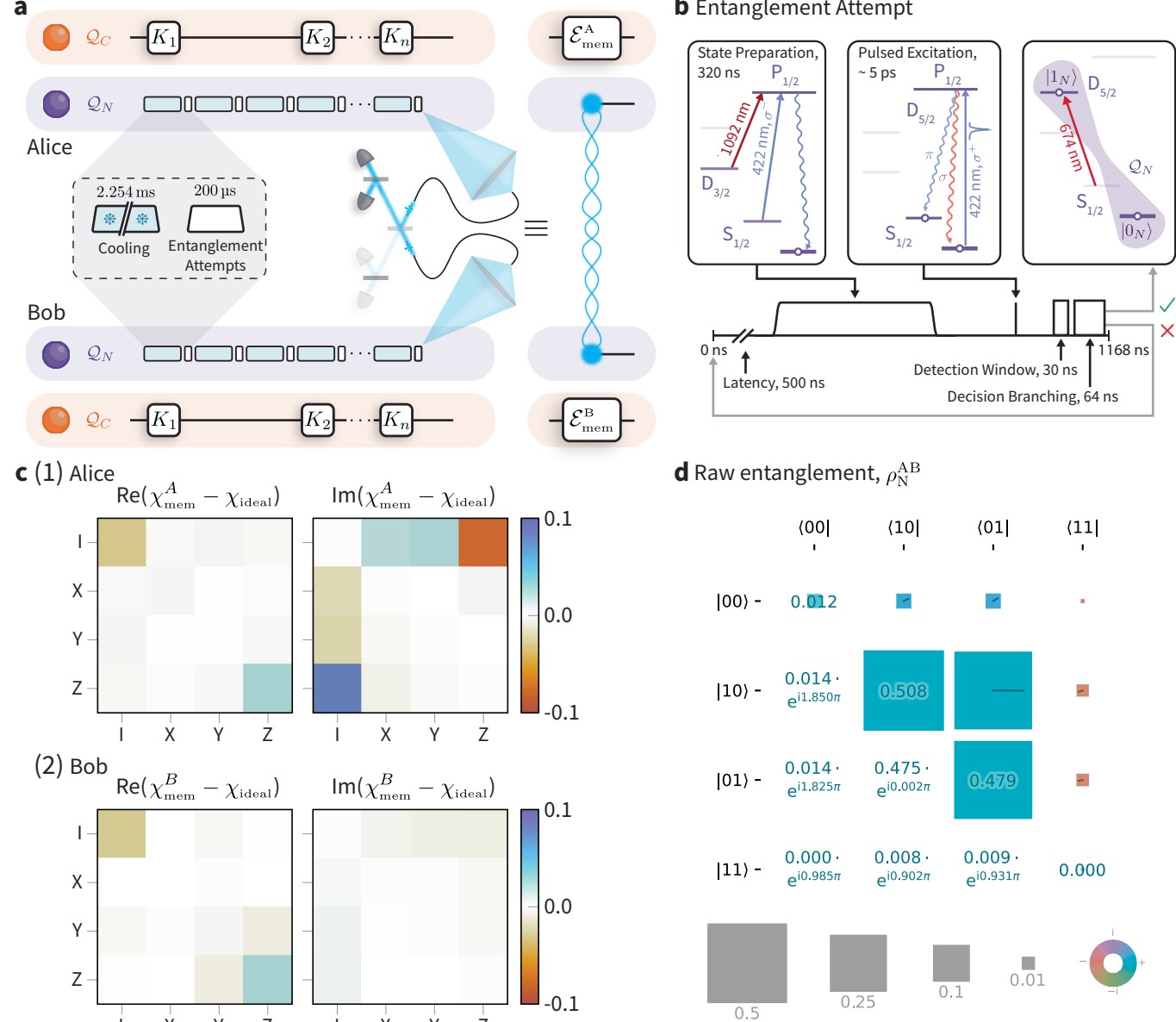

**c (1) Alice**

$$\text{Re}(\chi^A_{\text{mem}} - \chi_{\text{ideal}}) \qquad \text{Im}(\chi^A_{\text{mem}} - \chi_{\text{ideal}})$$

**(2) Bob**

$$\text{Re}(\chi^B_{\text{mem}} - \chi_{\text{ideal}}) \qquad \text{Im}(\chi^B_{\text{mem}} - \chi_{\text{ideal}})$$

**d** Raw entanglement, $\rho^{\text{AB}}_{\text{N}}$

**Extended Data Fig. 2 | Generation of remote entanglement and robust memory of the circuit qubits. a**, Entanglement is generated between the network qubits using 200 μs of entanglement attempts interleaved with 2.254 ms of sympathetic recooling using the Sr⁺ ion. This is repeated until the entanglement is successfully heralded by a particular detector click pattern. While attempting to generate entanglement between the network qubits, Knill dynamical decoupling pulses, $K_i$, are used to preserve the state of the circuit qubits. **b**, Each entanglement attempt has a total duration of 1168 ns. We perform a 320-ns state-preparation pulse (which has a switching latency of 500 ns), pumping the Sr⁺ ion into the lower ground Zeeman state. An approximately 5-ps pulse excites the Sr⁺ ion to the upper $P_{1/2}$ level (lifetime about 7 ns), which rapidly decays to one of the ground Zeeman levels, thereby generating ion–photon entanglement. We collect a photon from each of the modules, interfere

them on a beam splitter and perform a projective measurement on the two-photon polarization state. Particular detector click patterns occurring within the detection window herald the successful generation of remote entanglement. We then exit the attempt loop and map the entanglement into the optical network qubits, $\mathcal{Q}_N$, with an extra π-pulse on the 674-nm transition. **c**, Difference between the reconstructed process matrices, $\chi_{\text{mem}}$, for the process of storing the state of the circuit qubit in (1) Alice and (2) Bob while generating entanglement on the network qubits and the ideal process matrix, $\chi_{\text{ideal}} = \text{diag}(1, 0, 0, 0)$. The reconstructed process matrices have fidelities 98.1(4)% and 98.2(5)% for Alice and Bob, respectively. **d**, Reconstructed density matrix of the remotely entangled network qubits. The state has a fidelity of 96.89(8)% to the $|\Psi^+\rangle$ Bell state.

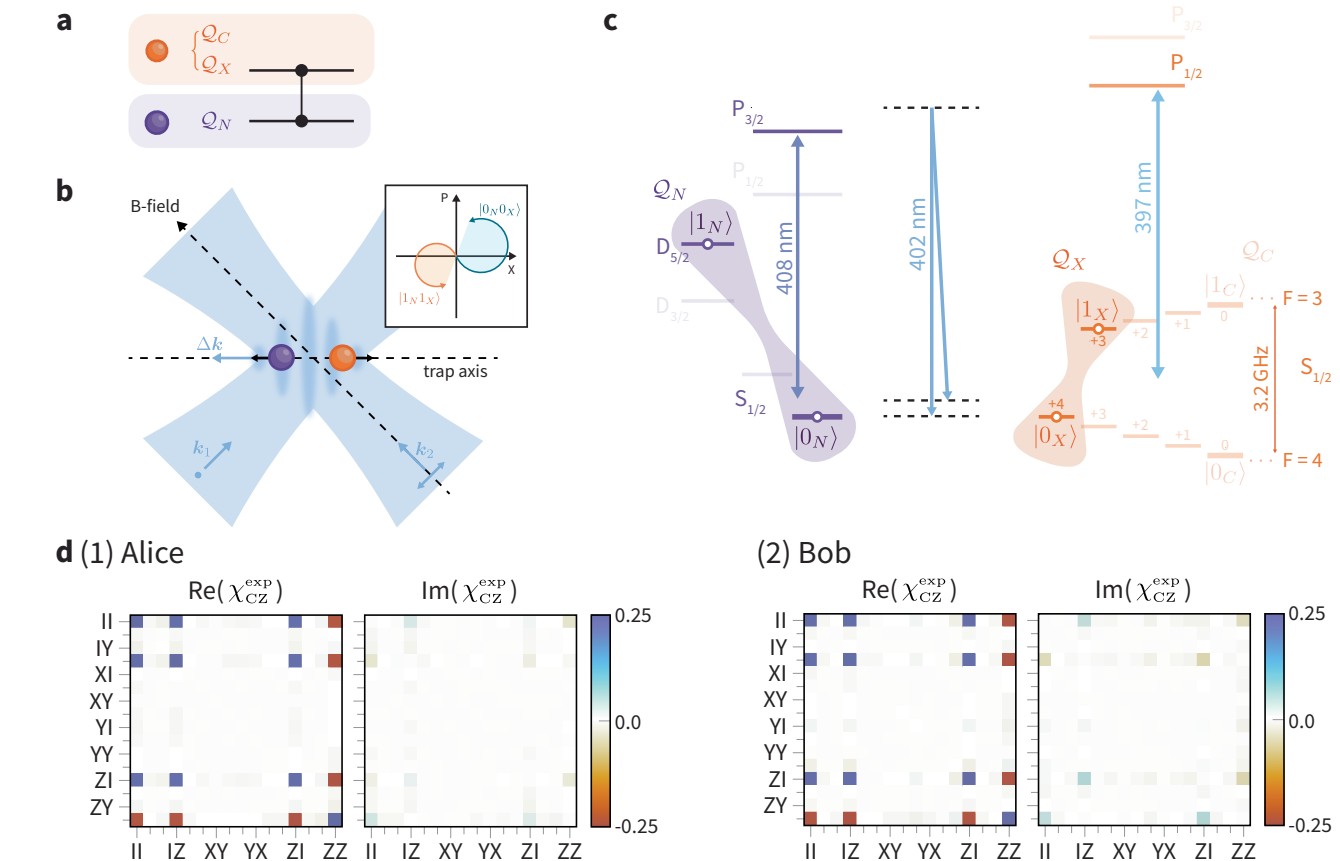

**Extended Data Fig. 3 | Implementation of the local mixed-species CZ gates.** **a**, Circuit element for the local CZ gate, implemented between the $\mathcal{Q}_N$ and $\mathcal{Q}_X$ qubits. **b**, Geometry for the mixed-species gate mechanism. A pair of Raman laser beams are aligned orthogonal to one another, such that their relative wavevector, $\Delta \mathbf{k}$ is along the trap axis. The interference of these beams leads to a polarization travelling-standing wave, which induces spin-dependent light shifts oscillating at a frequency close to the frequency of the axial out-of-phase motional mode. Owing to the spatial gradient of the light shift, the ions experience a spin-dependent force that displaces the spin states in phase space, as shown in the inset, thus enabling the implementation of geometric phase gates. **c**, Energy-level diagram for the gate mechanism acting on the $\mathcal{Q}_N$ and $\mathcal{Q}_X$ qubits. By tuning the Raman lasers to 402 nm, we couple to both the 397-nm $S_{1/2} \leftrightarrow P_{1/2}$ dipole transition in Ca$^+$ and the 408-nm $S_{1/2} \leftrightarrow P_{3/2}$ dipole transition in Sr$^+$. **d**, Process matrices for the local, mixed-species CZ gates for (1) Alice and (2) Bob. The process matrices have average gate fidelities of 97.6(2)% and 98.0(2)% for Alice and Bob, respectively.

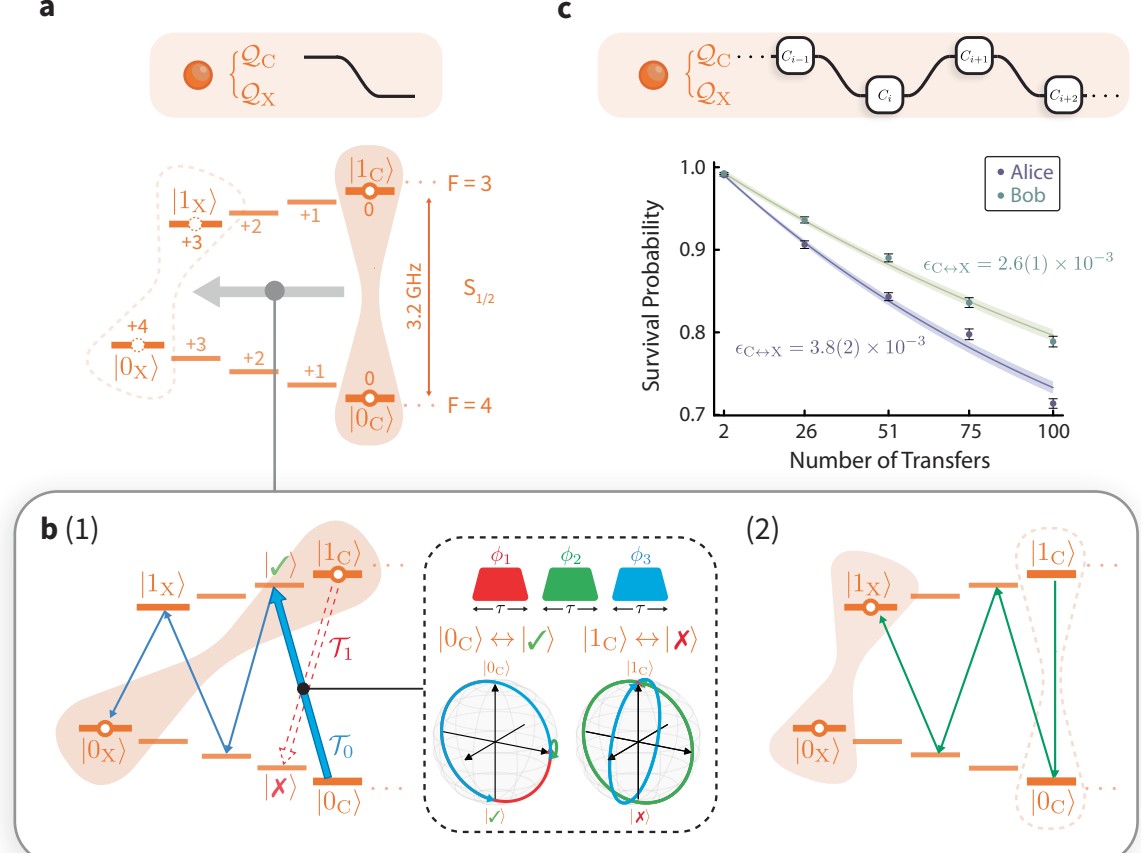

**Extended Data Fig. 4 | Transfer between the circuit and auxiliary qubits.**
**a**, Circuit element and level diagram showing the coherent transfer of quantum information from the $\mathcal{Q}_C$ qubit to the $\mathcal{Q}_X$ qubit. The inverse transfer is implemented by performing the same steps in reverse. **b**, The transfer pulse sequence comprises two steps. (1) The first step maps the state $|0_C\rangle$ to $|0_X\rangle$. Owing to the near-degeneracy of the intended transition $\mathcal{T}_0$: $|0_C\rangle \leftrightarrow |\checkmark\rangle$ (thick blue arrow) and the unwanted transition $\mathcal{T}_1$: $|1_C\rangle \leftrightarrow |X\rangle$ (red dashed arrow), separated by only about 15 kHz, we use a composite pulse sequence to suppress off-resonant coupling to the $\mathcal{T}_1$ transition. The composite pulse sequence, shown in the dashed box, comprises three pulses of duration $\tau$ resonant with the $\mathcal{T}_0$ transition with differing phases $\phi_i$. The pulse duration, $\tau$, is equal to the $2\pi$ time of the $\mathcal{T}_1$ transition, $\phi_1 = \phi_3 = 0$, and $\phi_2 \approx 2\pi \times 0.231$ is optimized

experimentally. The subsequent transfer pulses (thin blue arrows) are $\pi$-pulses on the relevant transitions. This sequence therefore performs the mapping $|0_C\rangle \rightarrow |0_X\rangle$, leaving the state $|1_C\rangle$ unaffected. (2) The second step comprises a sequence of $\pi$-pulses that maps $|1_C\rangle \rightarrow |1_X\rangle$. This completes the coherent transfer $\mathcal{Q}_C \rightarrow \mathcal{Q}_X$. **c**, The performance of the transfer sequence is characterized using a modified version of RBM, in which we alternately perform Clifford operations on the $\mathcal{Q}_C$ and $\mathcal{Q}_X$ qubits. By measuring the survival probability for different numbers of transfers, and neglecting the errors of the single-qubit gates $C_i$ (which are about $1 \times 10^{-4}$), we extract the error per transfer, $\epsilon_{C\leftrightarrow X}$, yielding $3.8(2) \times 10^{-3}$ and $2.6(1) \times 10^{-3}$ for Alice and Bob, respectively. All error bars indicate one standard deviation.

**Extended Data Table 1 | State-preparation and measurement errors for all of the qubit states, in each module**

| Module | Qubit | State-preparation $\epsilon(\times 10^{-3})$ | Readout $\epsilon_0(\times 10^{-3})$ | $\epsilon_1(\times 10^{-3})$ |
|--------|-------|-------------------|-------------------|-------------------|
| | $\mathcal{Q}_N$ | 4.7(5) | < 0.001 | 1.068(4) |
| Alice | $\mathcal{Q}_C$ | 4.1(3) | 4.1(4) | 2.7(4) |
| | $\mathcal{Q}_X$ | 3.1(4) | 1.77(4) | 0.357(4) |
| | $\mathcal{Q}_N$ | 5.0(5) | < 0.001 | 1.085(4) |
| Bob | $\mathcal{Q}_C$ | 4.6(3) | 2.7(5) | 1.7(5) |
| | $\mathcal{Q}_X$ | 3.8(4) | 1.02(2) | < 0.001 |

The average state-preparation and measurement error is 5.0(2)×10⁻³.