## [Peer Review File · Nature]

Distributed Quantum Computing across an Optical Network Link

Corresponding Author: Mr Dougal Main

Version 0:

Reviewer comments:

Referee #1

(Remarks to the Author)

The manuscript describes experiments carried out with two pairs of trapped ions. Each pair consists of a strontium ion ("network qubit") and a calcium ion ("circuit/auxiliary qubit") confined in a surface Paul trap under ultra-high vacuum. The two vacuum chambers are separated by 2 m and connected via an optical fiber link. Using this elementary quantum network, the authors (1) teleport a controlled-Z gate between circuit qubits deterministically and (2) implement an iSWAP gate, a SWAP gate, and Grover's search algorithm.

The larger context for this work is the challenge of scaling up quantum computers so that they can be useful, that is, dramatically increasing the number of qubits while achieving fidelities compatible with fault tolerance. Distributed quantum computing (DQC) via quantum gate teleportation (QGT) is a promising route to address this challenge, as the authors outline convincingly. The results presented here are, in my opinion, the most advanced demonstration of DQC to date and well suited for publication in Nature. In particular, this is the first demonstration of deterministic QGT, which represents an original and highly significant advance with respect to previous work. Moreover, the implementation of gates and algorithms requiring two and three instances of QGT is an important achievement enabled both by the high fidelity of the trapped-ion operations and the memory capabilities and spectral isolation of mixed-species nodes.

The manuscript is written in a clear, precise style and is easy to follow. The abstract and introduction are aimed appropriately at a nonspecialist audience and situate the results clearly with respect to the state of the art. There were several technical questions that came to my mind while reading the main text, but I found them answered in the Methods section, which I believe is the right place for that information. The results are communicated in a straightforward way, including nice visual representations in the figures. The methodology is valid and well motivated, and I have no concerns about the quality of the data. I particularly appreciated the quantitative discussion of the circuit performance (lines 249 to 276). The conclusions appear robust and valid and synthesize several established ideas for how to take the next steps to more complex quantum network architectures.

Here are some minor points that I ask the authors to take into consideration:

- In the first sentence of the abstract, the phrase "without compromising on performance and connectivity" is imprecise and perhaps closer to advertising text than scientific writing. It is likely that one will always have to make compromises of some sort in scaling up quantum computers (e.g., accept lower gate fidelities than would be possible with just two qubits). I understand the authors' point that all-to-all connectivity and high fidelities are possible in a DQC architecture, but those features do come with their own costs (e.g., added complexity).
- In the last sentence of the abstract, it is unclear what the phrase "this technique" refers to.
- It would be useful for the authors to address the earlier work of P. Maunz et al., Phys. Rev. Lett. 102, 250502 (2009) in their review of the state of the art (lines 89-120), as it is (to my knowledge) the earliest demonstration of a non-local entangling gate between trapped ions in separate modules, although it is heralded rather than deterministic.
- Average gate fidelity is stated multiple times in the manuscript. What does the uncertainty on the average gate fidelity represent? Let's say that I average over a tomographically complete set of four initial states, but the gate fidelities of the four cases differ from one another by more than the uncertainty associated with each fidelity (determined from resampling). It seems like both the individual uncertainties and the variation across initial states should be conveyed. I couldn't find a clear answer in the "Quantum process tomography" section of the methods (line 351).
- Distributed Quantum Computing (starting on line 201): It might be helpful to provide references to non-distributed versions

of the iSWAP and SWAP gates and Grover's algorithm, so that the reader could compare, e.g., fidelities.

- Line 329: The phrase "to provide a quantisation axis" is misleading (although it's common shorthand in the field): we always have the freedom to choose a quantisation axis, and we don't need a magnetic field to provide it. The point of the magnetic field is to lift the degeneracy of the Zeeman states in a known way.

Referee #2

(Remarks to the Author)

In their manuscript, Main et al. present the experimental implementation of distributed quantum computing. For this, two elementary quantum computing modules based on ions are connected together through a photonic link. Each module is made of two co-trapped ions: the Sr⁺ is used for the optical interface whereas the Ca⁺ is used for the computation tasks. To operate as a whole, the modules are connected together by entangling the Sr⁺ ions. This is done by a Bell states measurement on photons emitted by each ion, a capability demonstrated by the same group in previous publications (ref 28). As a CZ gate can be performed between the Sr⁺ and Ca⁺ ions, the entanglement can be shared with the "computation" ions too. It is worth noting that for this latter two bases are used, one for gate operations the other one for long-term storage of the qubits. This is quite essential as the entanglement via the optical link requires multiple trials until success and therefore the qubit carried by the Ca⁺ ion needs to be properly stored during this time. With this resource in hand the authors first investigate the teleportation of a controlled-Z gate and used the standard characterization method of process tomography. They then build up on that gate to demonstrate iSWAP and SWAP gate (that requires multiple of the CZ gate). Also, the authors could run an elementary Grover algorithm.

The manuscript is well written, quite easy to understand and the results are convincing. This is arguably an important step in quantum information: it does not set any new benchmark but an important milestone in the realization of scalable architecture of quantum computing systems. Therefore, I do believe this manuscript is suitable for publication in Nature.

I would however strongly encourage the authors to consider the following remarks to improve their manuscript:

- 1) Why aren't there more Ca⁺ ions as depicted in figure 1 a? because this would have been more in the spirit of "distributed quantum computing", here the individual modules cannot perform any quantum computation so it is a bit questionable to say that the computation is distributed.
- 2) Despite the impressive fidelity of the Bell state (97%) I could not find a discussion on the potential sources of error.
- 3) There isn't much about the photonic part, and some information are not even in ref 28. E.g. the wavelength of the photons requires a bit of digging to find it. How short/long are the photons? Any histogram? What is the indistinguishability? HOM? Any form of postselection on the detection events? How does this connect with the Bell state fidelity?
- 4) Why is the link (only) 2 meters? Could it be longer e.g. 10m? 100m?
- 5) (more on the cosmetic side) Despite the good readability of the manuscript, I think there is a little overuse of acronyms, this makes the reading a bit annoying sometimes.

Response to Referees

October 29, 2024

Dear Referees,

We thank the referees for their time and detailed reading of the paper. We have modified our manuscript accordingly and believe the changes have improved the paper. Our responses are provided below. Additionally, we attach two documents: one with changes highlighted, and another with the changes fully incorporated into the manuscript. Please note that the line numbers given below correspond to the position of the changes in the manuscript that has the highlighted changes.

Referee 1 Comments

*The manuscript describes experiments carried out with two pairs of trapped ions. Each pair consists of a strontium ion (“network qubit”) and a calcium ion (“circuit/auxiliary qubit”) confined in a surface Paul trap under ultra-high vacuum. The two vacuum chambers are separated by 2 m and connected via an optical fiber link. Using this elementary quantum network, the authors (1) teleport a controlled-Z gate between circuit qubits deterministically and (2) implement an *i*SWAP gate, a SWAP gate, and Grover’s search algorithm.*

*The larger context for this work is the challenge of scaling up quantum computers so that they can be useful, that is, dramatically increasing the number of qubits while achieving fidelities compatible with fault tolerance. Distributed quantum computing (DQC) via quantum gate teleportation (QGT) is a promising route to address this challenge, as the authors outline convincingly. The results presented here are, in my opinion, the most advanced demonstration of DQC to date and well suited for publication in *Nature*. In particular, this is the first demonstration of deterministic QGT, which represents an original and highly significant advance with respect to previous work. Moreover, the implementation of gates and algorithms requiring two and three instances of QGT is an important achievement enabled both by the high fidelity of the trapped-ion operations and the memory capabilities and spectral isolation of mixed-species nodes.*

The manuscript is written in a clear, precise style and is easy to follow. The abstract and introduction are aimed appropriately at a nonspecialist audience and situate the results clearly with respect to the state of the art. There were several technical questions that came to my mind while reading the main text, but I found them answered in the Methods section, which I believe is the right place for that information. The results are communicated in a straightforward way, including nice visual representations in the figures. The methodology is valid and well motivated, and I have no concerns about the

quality of the data. I particularly appreciated the quantitative discussion of the circuit performance (lines 249 to 276). The conclusions appear robust and valid and synthesize several established ideas for how to take the next steps to more complex quantum network architectures.

Here are some minor points that I ask the authors to take into consideration:

We thank referee 1 for comments on the manuscript. Responses to the comments made by referee 1 are provided below. Please note that changes to the manuscript text in response to comments made by referee 1 are **highlighted in red**.

- **Referee 1, Comment 1:** *In the first sentence of the abstract, the phrase “without compromising on performance and connectivity” is imprecise and perhaps closer to advertising text than scientific writing. It is likely that one will always have to make compromises of some sort in scaling up quantum computers (e.g., accept lower gate fidelities than would be possible with just two qubits). I understand the authors’ point that all-to-all connectivity and high fidelities are possible in a DQC architecture, but those features do come with their own costs (e.g., added complexity).*

Response: We agree that this phrase may be a little strong. The intent of this phrase is to convey that an ideal distributed quantum computing architecture would provide a user with the same capabilities as an ideal non-distributed architecture of the same size. Of course, as the referee mentions, in a practical setting, compromises will always be needed. We have modified the text to emphasise that we are referring to the goal of the ideal distributed quantum computing architecture, rather than implying that this architecture provides some kind of “free lunch”.

The first sentence of the abstract now reads: “Distributed quantum computing (DQC) combines the computing power of multiple networked quantum processing modules, **ideally** enabling the execution of large quantum circuits...”

- **Referee 1, Comment 2:** *In the last sentence of the abstract, it is unclear what the phrase “this technique” refers to.*

Response: We have amended the manuscript so that it is clear that “this technique” refers to the distributed quantum computing architecture.

The last sentence of the abstract now reads: “As photons can be interfaced with a variety of systems, **the versatile DQC architecture demonstrated here provides a viable pathway towards large-scale quantum computing for a range of physical platforms.**~~this technique has applications extending beyond trapped-ion quantum computers, providing a viable pathway towards large-scale quantum computing for a range of physical platforms.~~”

- **Referee 1, Comment 3:** *It would be useful for the authors to address the earlier work of P. Maunz et al., Phys. Rev. Lett. 102, 250502 (2009) in their review of the state of the art (lines 89-120), as it is (to my knowledge) the earliest demonstration of a non-local entangling gate between trapped ions in separate modules, although it is heralded rather than deterministic.*

Response: In the work by P. Maunz et al., a non-local entangling gate is performed between two ytterbium ions situated in separate traps interconnected

by an optical quantum network. This experiment is similar to the work by Daiss *et al.*, that is cited in the manuscript (ref. 35), in that the non-local gate is realised by mediating an interaction between a qubit and a photon, followed by the direct transfer of the photon across the optical network (in the case of Daiss *et al.*, this photon is sent to interact with the qubit in the second network module, while in the case of Maunz *et al.*, photons from each module are interfered at a central heralding station). Both experiments make use of what we referred to as “direct transfer” in the manuscript; in these schemes, loss of the transferred photonic qubit directly results in an unrecoverable loss in quantum information. As noted by the referee, these gates are heralded processes and inherently non-deterministic. This is in contrast to the quantum gate teleportation scheme implemented in our work, in which the optical network is exclusively used to generate remote entanglement between the network modules.

We agree that this is an important result that should be included in the review of the state-of-the-art, and have amended the manuscript appropriately.

The revised text (lines 113-123) is: “~~Additionally, there have been demonstrations of heralded non-local entangling gates across a photonic quantum network in which photons are used to directly transfer quantum information between modules^{35,36}. Daiss *et al.*³⁶ demonstrated a heralded non-local entangling gate across a photonic quantum network using a photon to directly transfer quantum information between modules.~~ However, in these demonstrations, unavoidable photon loss destroyed ~~photon loss necessarily destroys~~ the states of the circuit qubits, rendering ~~these~~ schemes non-deterministic. ”

- **Referee 1, Comment 4:** *Average gate fidelity is stated multiple times in the manuscript. What does the uncertainty on the average gate fidelity represent? Let’s say that I average over a tomographically complete set of four initial states, but the gate fidelities of the four cases differ from one another by more than the uncertainty associated with each fidelity (determined from resampling). It seems like both the individual uncertainties and the variation across initial states should be conveyed. I couldn’t find a clear answer in the “Quantum process tomography” section of the methods (line 351).*

Response: The process matrix provides a complete description of a process, allowing us to predict the output state for any given input state. The average gate fidelity metric is used to compare the performance of the reconstructed quantum process to an ideal unitary operation, and is defined in Nielsen [1]. In this definition, the average gate fidelity is calculated as the average, over all pure input states, of the fidelity between the actual output state produced by the process \mathcal{E} and the ideal output state produced by the unitary operation U . This is a useful metric, since the input states are generally unknown, and part of a longer-running distributed quantum computation. We have now included a definition of the average gate fidelity in the “Quantum process tomography” section of the Methods.

The uncertainty on the average gate fidelity is quoted as one standard deviation of the average gate fidelities calculated from processes reconstructed using resampled data sets obtained via bootstrapping. This uncertainty represents the sensitivity of the analysis to statistical fluctuations in the input data. Crucially, in the limit of an infinite number of tomographic measurements, the statistical

uncertainty on the value of the average gate fidelity will tend to zero, since we would be able to perfectly reconstruct the process.

By construction, the average gate fidelity is a single parameter summarising the overall performance of a process, and does not capture the variation of output state fidelities across different input states. This variation in output state fidelities is inherent to the process and is related to the structure of the process matrix, but it is not represented by either the average gate fidelity or its uncertainty. To illustrate how the structure of the process matrix relates to variations in the fidelities of different input states, we provide two examples of processes that exhibit the same average gate fidelity but different distributions of output state fidelities.

The first example is a dephasing channel, given by

$$\mathcal{E}(\hat{\rho}) = (1 - p)\hat{\rho} + p\hat{\sigma}_z\hat{\rho}\hat{\sigma}_z, \quad (1)$$

for some parameter p . This process leaves the eigenstates of $\hat{\sigma}_z$, i.e., the computational basis states, unchanged; however, it causes dephasing of superposition states such as the eigenstates of $\hat{\sigma}_x$. Consequently, the fidelities for different input states range from 1 (for $\hat{\sigma}_z$ eigenstates) to $1 - p$ (for $\hat{\sigma}_x$ and $\hat{\sigma}_y$ eigenstates), resulting in an average gate fidelity of $1 - \frac{2p}{3}$. The second example is a depolarising channel, given by

$$\mathcal{E}(\hat{\rho}) = (1 - p)\hat{\rho} + \sum_{i=x,y,z} \frac{p}{3}\hat{\sigma}_i\hat{\rho}\hat{\sigma}_i. \quad (2)$$

This process affects all states equally, and the output state fidelity for each input state is $1 - \frac{2p}{3}$. As a result, there will be no spread in the output state fidelities.

In summary, the average gate fidelity metric quantifies the performance of a process for unknown input states, and is thus defined as the average output state fidelity over all possible pure input states. The uncertainty quoted for the average gate fidelity represents the statistical uncertainty in our experimental determination of this metric, obtained from resampling. Since the average gate fidelity is a single parameter summarising the overall performance, it does not capture the variation of output state fidelities across different input states. Therefore, the spread of output fidelities is not reflected in either the average gate fidelity or its uncertainty. It is possible to calculate the spread of output state fidelities for any set of input states using the process matrix shown in Fig 2c. However, we elected not to include this in the manuscript as we believe it does not aid the understanding of the results; we can provide this data upon reasonable request.

To clarify the definition of the average gate fidelity and the way we quote its uncertainty, we have modified the manuscript as follows. We have added the definition of the average gate fidelity (lines 415-424): “ **To quantify the performance of a process, \mathcal{E} , compared to an ideal unitary process, U , we make use of the average gate fidelity**

$$\bar{F}_{\mathcal{E},U} = \int d\psi \langle \psi | U^\dagger \mathcal{E}(|\psi\rangle\langle\psi|) U |\psi\rangle \quad (3)$$

as defined by Nielsen⁵⁹, which corresponds to the fidelity averaged over all pure input states. We define the process \mathcal{E}' as the application of the process \mathcal{E} , followed by the inverse of the ideal process U , such that

$$\mathcal{E}'(\rho) = U^\dagger \mathcal{E}(\rho) U. \quad (4)$$

If $\chi'_{\alpha\beta}$ is the process matrix representing \mathcal{E}' , as in (1), then the average gate fidelity can be expressed as

$$\bar{F}_{\mathcal{E},U} = \frac{1 + d\chi'_{00}}{1 + d}, \quad (5)$$

where d is the dimension of the Hilbert space. ”

Note that equation (1) now refers to the (previously unlabelled) definition of the process matrix representation at the start of the “Quantum process tomography” section.

Additionally, we have made it clear in the discussion of resampling, that the error bars are quoted as the standard deviation of the spread of average gate fidelities from processes reconstructed from resampled data sets. The revised text (line 429-435) is: “ ~~The error bar on the average gate fidelity of a reconstructed process is quoted as the standard deviation of average gate fidelities of processes reconstructed from resampled data sets. Error bars on the fidelities of reconstructed processes are quoted as the standard deviation of the fidelities of the resampled data sets.~~ ”

- **Referee 1, Comment 5:** *Distributed Quantum Computing (starting on line 201): It might be helpful to provide references to non-distributed versions of the *i*SWAP and SWAP gates and Grover’s algorithm, so that the reader could compare, e.g., fidelities.*

Response: We agree that this comparison would add value to the manuscript. In particular, we believe that the comparison to the early implementations of quantum algorithms would be of interest to the reader. To that end, we have added citations for two of the original (non-distributed) demonstrations of Grover’s algorithm with two qubits, one implemented using trapped-ions [Brickman *et al.*, *Phys. Rev. A* **72**, 050306 (2005)] and one using superconducting qubits [DiCarlo *et al.*, *Nature* **460**, 240–244 (2009)]. The performance of the non-distributed implementations of Grover’s algorithm reported in these references is comparable to the performance reported here. Over the almost two decades since these results were reported, we have seen significant advances in quantum information processing technologies, and thus this comparison hints at the possible future performance of distributed quantum computation.

The revised text (line 231) is: “Finally, we implement Grover’s algorithm^{5,41,42} ⁵ on our distributed quantum processor. ”

- **Referee 1, Comment 6:** *Line 329: The phrase “to provide a quantisation axis” is misleading (although it’s common shorthand in the field): we always have the freedom to choose a quantisation axis, and we don’t need a magnetic field to provide it. The point of the magnetic field is to lift the degeneracy of the Zeeman states in a known way.*

Response: While it is important that the magnetic field lifts the degeneracies of the Zeeman states, this static field also defines a direction in space to which we reference various optical field polarisations. This is particularly important for our ion-photon interface, which relies on the polarisation encoding of the photonic qubits. We have adjusted the manuscript to clarify the role of the applied magnetic field.

The revised text (lines 346-348) is: “A ~ 0.5 mT magnetic field is applied parallel to the surface of the trap to **lift the degeneracies of the Zeeman states** and provide a quantisation axis.”

Referee 2 Comments

*In their manuscript, Main et al. present the experimental implementation of distributed quantum computing. For this, two elementary quantum computing modules based on ions are connected together through a photonic link. Each module is made of two co-trapped ions: the Sr^+ is used for the optical interface whereas the Ca^+ is used for the computation tasks. To operate as a whole, the modules are connected together by entangling the Sr^+ ions. This is done by a Bell states measurement on photons emitted by each ion, a capability demonstrated by the same group in previous publications (ref 28). As a CZ gate can be performed between the Sr^+ and Ca^+ ions, the entanglement can be shared with the “computation” ions too. It is worth noting that for this latter two bases are used, one for gate operations the other one for long-term storage of the qubits. This is quite essential as the entanglement via the optical link requires multiple trials until success and therefore the qubit carried by the Ca^+ ion needs to be properly stored during this time. With this resource in hand the authors first investigate the teleportation of a controlled-Z gate and used the standard characterization method of process tomography. They then build up on that gate to demonstrate *i*SWAP and SWAP gate (that requires multiple of the CZ gate). Also, the authors could run an elementary Grover algorithm.*

The manuscript is well written, quite easy to understand and the results are convincing. This is arguably an important step in quantum information: it does not set any new benchmark but an important milestone in the realization of scalable architecture of quantum computing systems. Therefore, I do believe this manuscript is suitable for publication in Nature.

I would however strongly encourage the authors to consider the following remarks to improve their manuscript:

We thank referee 2 for comments on the manuscript. Responses to the comments made by referee 2 are provided below. Please note that changes to the manuscript text in response to comments made by referee 2 are **highlighted in blue**.

- **Referee 2, Comment 1:** *Why aren't there more Ca^+ ions as depicted in figure 1 a? because this would have been more in the spirit of “distributed quantum computing”, here the individual modules cannot perform any quantum computation so it is a bit questionable to say that the computation is distributed.*

Response: We disagree that “it is a bit questionable to say that the computation is distributed”. Each module in our setup contains one network qubit and one circuit qubit, which constitutes the minimum requirement for participating in distributed quantum computation. While it is true, as the referee points out, that a single circuit qubit in isolation cannot perform meaningful quantum computation, this does not invalidate the claim of distributed quantum computing in our setup. For example, one could in principle construct an N -qubit distributed quantum computer by connecting N modules, each containing one network qubit and one circuit qubit.

We believe the confusion arises from the analogy to distributed classical computing. In distributed classical computing, the computation is distributed by assigning separate tasks or instructions to different modules, and a single bit per module would indeed not suffice. However, in distributed quantum computing, the key resource being distributed is entanglement between the modules, not computational instructions. This entanglement enables the modules to operate collectively as a fully intra-connected quantum computer, even if individual modules cannot perform meaningful quantum computation on their own.

In summary, the aim of our work is the proof-of-principle demonstration of distributed quantum computing, and hence we implement the simplest configuration: two modules each containing one network qubit and one circuit qubit. This is the minimum requirement for demonstration universal distributed quantum computing. To avoid potential misunderstandings, we have made some small adjustments to the manuscript to emphasise that a distributed quantum computer can be partitioned into modules with at least one network qubit and one circuit qubit.

We have revised the caption for Figure 1 with the following: “The modules consist of at least one ~~small number of~~ network qubits (purple) and at least one circuit qubits (orange), which may directly interact via local operations.”

Additionally, we have added the sentence (lines 84-87): “Quantum circuits can be partitioned freely in this architecture, down to a minimum of one circuit qubit per module in the fully distributed case.”

Finally, Fig 1 has been adjusted to show modules containing different numbers of network and circuit qubits, with at least one network and one circuit qubit in each module.

- **Referee 2, Comment 2:** *Despite the impressive fidelity of the Bell state (97%) I could not find a discussion on the potential sources of error.*

Response: The remote entanglement infidelity ($\sim 3\%$) is consistent with the ion-photon entanglement infidelity ($\sim 1.5\%$ from each module). We therefore attribute the majority of the error to the process by which we generate ion-photon entanglement in each module. Specifically, we believe that the dominant contributions to the infidelity arise from imperfections in the optical systems used to collect the single photons from each module, as well as drifts in the birefringence of the optical fibres used to link the modules. Other sources of error, such as photon distinguishability in the Bell state measurement due to, e.g., imperfect overlap on the beamsplitter, are believed to contribute less significantly, and at a level too small for us to measure easily.

We have added the following to the manuscript: “We believe that the fidelity is predominantly limited by errors occurring during the generation of ion-photon entanglement in each module, rather than imperfections in the apparatus used to perform the projective Bell state measurement. In particular, we attribute the primary sources of error to polarisation mixing due to imperfections in the imaging systems used to collect single photons from each module, and to drifts in the birefringence of the optical fibres that form the network link between the modules.”

- **Referee 2, Comment 3:** *There isn't much about the photonic part, and some information are not even in ref 28.*

Response: We agree that there is not much information on the photonic part in this paper, or in ref 28, as pointed out by the referee, but it can be found in the (open access) DPhil theses of Stephenson [2] and Nadlinger [3]. We have therefore added citations to these theses into the manuscript (line 445), so that the interested reader may find more information about our photonic interface.

E.g. the wavelength of the photons requires a bit of digging to find it. How short/long are the photons?

Response: The photons are generated by spontaneous emission from the excited $5P_{1/2}$ manifold to the ground $5S_{1/2}$ manifold in $^{88}\text{Sr}^+$. As a result, the wavelength of the spontaneously emitted photons is 422 nm, and the temporal profile of the photons is an exponential decay with a time constant given by the lifetime of the $5P_{1/2}$ manifold (7.4 ns).

We have revised the text (line 440) as follows: “Spontaneously-emitted 422 nm photons entangled with the Sr^+ ions are ...”

Any histogram? What is the indistinguishability? HOM?

Response: For the Bell state measurement, it is crucial that the photons from each module are indistinguishable when they interfere at the non-polarising beam-splitter (NPBS). This therefore requires that the photons arrive at the NPBS at the same time. To achieve a remote entanglement fidelity error of $< 1\%$, the timing mismatch needs to be < 150 ps, which corresponds to a free space path length mismatch of < 4.5 cm [3]. We can tune the photon arrival timings by adjusting the free space path length for the picosecond pulse that excites the Sr^+ ions to the upper $P_{1/2}$ manifold. We have previously observed timing mismatches of as little as 13 ps [2], which would contribute a remote entanglement error of $< 10^{-3}$.

We have extensive theoretical analysis of the imperfections of the Bell state measurement [3]; however, we have not yet performed a full experimental characterisation. As we mention in the response to Referee 2 - Comment 2, the fidelity of our remote entanglement is consistent with the fidelity of our ion-photon entanglement, and thus we believe that the error arising from the distinguishability of the photons is significantly smaller than the errors arising in the generation of ion-photon entanglement. As we do not believe it is a limiting mechanism, we have not performed a characterisation of the distinguishability of the photons, such as observing a HOM dip.

Any form of postselection on the detection events? How does this connect with the Bell state fidelity?

Response: Our experiment does not make any use of post-selection.

- **Referee 2, Comment 4:** *Why is the link (only) 2 meters? Could it be longer e.g. 10m? 100m?*

Response: The optical path length of the link is approximately 4 m. The line-of-sight distance between the two modules in the laboratory is approximately 2 m. In principle, there is no fundamental limit to the length of this optical link.

The main technical limitation is the exponential reduction in the entanglement generation success rate due to photon loss; at the 422 nm wavelength used in this work, optical fibres typically exhibit an attenuation of $\sim 33 \text{ dB km}^{-1}$.

In the context of distributed quantum computing, the shorter, metre-scale link (as in this work) is relevant for a “server room” architecture, where one could envisage a quantum computer comprising a large number of modules within a single room/building that are interconnected via optical network links. Of course, the ability to interconnect quantum processors over larger distance would enable the creation of a so-called “quantum internet”. Over these larger distances, using 422 nm photons (as in this work) would likely be extremely inhibiting due to the significant photon loss at this wavelength. However, by either utilising down-conversion or a different choice in optical transition used to realise the interface, it would be possible to operate using infrared photons where the fibre attenuation is significantly lower, and thus the interconnection of quantum processors over, e.g., metropolitan distances could be achieved. Furthermore, interconnection over, e.g., international distances could be achieved by utilising quantum repeater technologies. While such techniques are extremely interesting and are being actively pursued by many research groups, this was not within the scope of this work.

- **Referee 2, Comment 5:** *(more on the cosmetic side) Despite the good readability of the manuscript, I think there is a little overuse of acronyms, this makes the reading a bit annoying sometimes.*

Response: We agree with this observation and have removed the acronyms for quantum process tomography (QPT) in the main text (the acronym is still used in the Methods section) and for continuous-variable quantum computing (CVQC).

Additional Changes

In addition to the changes made in response to these comments, we have made some further changes of our own, all intended to improve the analysis and/or clarity of the manuscript. These changes are **highlighted in green**, and are discussed below.

The most significant of these changes is in regards to the modelling of the state-preparation and measurement errors in the tomography procedures. We now model the state-preparation and measurement imperfections separately, rather than combining them into a single error. Additionally, we consider only imperfections in the state-preparation and measurement processes themselves, rather than the additional error associated with the single-qubit rotations, R_i , since we do not believe we have an adequate model for these errors. We believe this new analysis is a more accurate model of the errors, however it does not significantly change any results presented in the manuscript.

The changes to the manuscript that are associated with this are listed below.

- Lines 392-413: “ ~~However, SPAM errors would manifest as errors in the reconstructed process; we therefore model these errors by replacing the σ_z measurement with~~

positive operator-valued measures (POVMs), where ϵ_0 (ϵ_1) is the SPAM error associated with the $|0\rangle$ ($|1\rangle$) qubit state. The values used for these operators are given in Ext. Fig. 2.

However, state-preparation and measurement errors would manifest as errors in the reconstructed process. We therefore model the imperfect state-preparation by replacing the ideal input states, $|\psi_i\rangle$, with the states

$$\rho_i = R_i [(1 - \epsilon) |0\rangle \langle 0| + \epsilon |1\rangle \langle 1|] R_i^\dagger,$$

where ϵ is the state-preparation error. Note that this model assumes that imperfect state preparation leaves the ionic state within the qubit subspace; however, imperfect state-preparation often results in leakage outside of this subspace. Nevertheless, for the purposes of our analysis, this model is sufficient.

Similarly, we model the imperfect qubit readout by replacing the projectors $P_{0,j}$ and $P_{1,j}$ with the positive operator-valued measures (POVMs),

$$\begin{aligned} M_{0,j} &= (1 - \epsilon_0) R_j^\dagger |0\rangle \langle 0| R_j + \epsilon_1 R_j^\dagger |1\rangle \langle 1| R_j \\ M_{1,j} &= (1 - \epsilon_1) R_j^\dagger |1\rangle \langle 1| R_j + \epsilon_0 R_j^\dagger |0\rangle \langle 0| R_j, \end{aligned}$$

where ϵ_0 and ϵ_1 are the computational basis readout errors. The values for these errors are given in Ext. Fig. 2. ”

- We have updated the manuscript with the values, density matrices, and process matrices obtained from state and process tomography, as calculated with this new analysis. In particular, the values that have changed are:
 - The teleported CZ gate fidelity: was 86.1(9) %, and is now 86.2(9) %.
 - The fidelity of the raw entanglement to the $|\Psi^+\rangle$ state: was 97.15(9) %, and is now 96.89(8) %.
 - The average gate fidelity of the local mixed-species CZ gate in Alice: was 97.5(2) %, and is now 97.6(2) %.
 - The total predicted error in the error budget (Table 1): was 11.9(6) %, and is now 12.1(6) %.
- Lines 481-483: “...reconstruction of the density matrix using the POVMs in (5) and (6).”
- Extended Data Figure 2: Rather than displaying the combined state-preparation and measurement errors, we now present the errors separately. The state-preparation errors are denoted ϵ , and the readout errors of the two computational basis states are denoted ϵ_0 and ϵ_1 . We have also changed the caption accordingly. Finally, we have formatted this extended data table according to the Nature guidelines (e.g., using 7pt font sans-serif font, etc...).

Finally, we have made some additional minor changes to the manuscript.

- First sentence of the abstract: “...the execution of large quantum circuits without compromising on performance or qubit and connectivity”

- Third sentence of the abstract: “. . . ; until now, ~~no demonstration has satisfied these requirements~~~~there has been no demonstration satisfying these requirements.~~”
- Fourth sentence of the abstract: “~~In this work, w~~We experimentally demonstrate. . .”
- Fifth sentence of the abstract: “The modules, ~~are~~ separated by ~ 2 m, ~~and~~ each contains dedicated network and circuit qubits.”
- Lines 9-11: “However, regardless of the physical platform used to realise ~~at~~the quantum computer. . .”
- Lines 53-55: “. . . , eliminating the need for post-selection of singular successful outcomes out of an exponentially large set of undesired ~~ones~~~~outcomes.~~”
- Lines 80-81: “This entanglement can then be used to mediate ~~multi-qubit~~entangling gates . . .”
- Lines 106-113: “In the trapped-ion QCCD architecture, Wan *et al.*²⁰ demonstrated QGT ~~between ions in two zones of the same trap, separated by ~ 840 μm ; the entanglement was deterministically generated between qubits via local operations before the qubits were transported~~~~in which the entanglement was deterministically generated between two “network” qubits via local operations before being transported ~ 840 μm to two separate locations within the same trap.~~”
- Lines 180-181: “This ~~entanglement is generated~~~~is done~~ via a try-until-success process, . . .”
- Lines 182-183: “~~In contrast to the network qubits, t~~The circuit qubits provide . . .”
- Lines 232-233: “This algorithm ~~searches~~~~considers~~ ~~searching~~ through a set of . . .”
- Caption of Figure 3: “. . . indicate ~~an~~ average gate fidelities~~y~~ of . . .”
- Lines 271-272: “The majority of identified errors~~sources~~ occur during . . .”
- Lines 272-274: “~~O~~~~It is worth noting that our~~ local errors do not represent the state-of-the-art ~~for~~trapped-ion processors; . . .”
- Lines 274-275: “. . . local operations exceeding the $\approx 99\%$ fidelity threshold . . .”
- Added three more references on line 277: “. . . in this ~~platform~~^{39,40,43-48}~~platform~~^{39,40,45,47,48}.”
- Line 280: “. . . with a gate ~~fidelity of 99.8(1)%~~~~error of 0.2(1)%.~~”
- Line 295: “With~~only~~ 3 circuit qubits . . .”
- Lines 354-356: “. . . ~ 2 orders of magnitude lower than that of the \mathcal{Q}_N qubit ~~with a sensitivity of -11.2 MHz mT^{-1} . . .”~~”
- Line 366: “We make use of this ~~property~~ for . . .”
- Line 388: “. . . rotating the output state $\mathcal{E}[\rho_i]$ by $R_j^\dagger R_j$. . .”
- Lines 453-454: “. . . , indicates the creation of the maximally entangled Ψ^+ Bell-state, . . .”

- Lines 867-869: “We thank Chris Ballance and Laurent Stephenson for design and construction of the apparatus, Jacob Blackmore and Péter Juhász for comments on the manuscript, . . .”
- Caption for Extended Data Figure 1: “. . . for both species; ions in the shelved state do not scatter photons, to indicate whether the ion is shelved.”
- Caption for Extended Data Figure 3: “. . . map the entanglement into the optical network qubits, \mathcal{Q}_{net} , with an additional π -pulse on the 674 nm transition.”
- Caption for Extended Data Figure 4: “Due to the spatial gradient of the light-shift, the ions ~~The ions therefore~~ experience a spin-dependent force . . .”
- Figure 2: We made the beamsplitter darker to improve visibility.
- Figure 3: We have added boxes in the distributed circuits to emphasise that the qubits belong to two separated modules. Additionally, we have aligned the base of figure c(ii) with the base of figure b(ii).
- Extended Data Figure 3: In (a), we have added labels for the two modules. In (c), we now plot the difference between the reconstructed storage process matrices, χ_{mem}^A and χ_{mem}^B , to the ideal storage process matrix, $\chi_{\text{ideal}} = \text{diag}(1, 0, 0, 0)$. We have adjusted the caption accordingly. Furthermore, we have removed extraneous tick marks from these matrices. In (d), we have added text to indicate the magnitude and phase of the density matrix elements.
- Extended Data Figure 4: We have changed the vector formatting to bold font, in accordance with the manuscript checklist.
- Extended Data Figure 5: Fixed two mistakes: in the inset in b(i), the trajectories on the left Bloch sphere were incorrectly coloured, and in b(ii) the circuit qubit now has a dotted outline, and the auxiliary qubit now has a solid fill.

We believe the changes improve the clarity and quality of the manuscript. Thank you for your thorough review, and we hope that the revised version meets your expectations.

Sincerely,
 Dougal Main (on behalf of the authors)

References

- [1] Nielsen, M. A. A simple formula for the average gate fidelity of a quantum dynamical operation. *Physics Letters A* **303**, 249 (2002).
- [2] Stephenson, L., *Entanglement between nodes of a quantum network*, DPhil Thesis, University of Oxford (2019).
- [3] Nadlinger, D. P., *Device-independent key distribution between trapped-ion quantum network nodes*, DPhil Thesis, University of Oxford (2022).